# Feature Selection in the Contrastive Analysis Setting

**Ethan Weinberger**
Paul G. Allen School of Computer Science
University of Washington
Seattle, WA 98195
`ewein@cs.washington.edu`

**Ian C. Covert**
Department of Computer Science
Stanford University
Stanford, CA 94305
`icovert@stanford.edu`

**Su-In Lee**
Paul G. Allen School of Computer Science
University of Washington
Seattle, WA 98195
`suinlee@cs.washington.edu`

## Abstract

Contrastive analysis (CA) refers to the exploration of variations uniquely enriched in a *target* dataset as compared to a corresponding *background* dataset generated from sources of variation that are irrelevant to a given task. For example, a biomedical data analyst may wish to find a small set of genes to use as a proxy for variations in genomic data only present among patients with a given disease (target) as opposed to healthy control subjects (background). However, as of yet the problem of feature selection in the CA setting has received little attention from the machine learning community. In this work we present contrastive feature selection (CFS), a method for performing feature selection in the CA setting. We motivate our approach with a novel information-theoretic analysis of representation learning in the CA setting, and we empirically validate CFS on a semi-synthetic dataset and four real-world biomedical datasets. We find that our method consistently outperforms previously proposed state-of-the-art supervised and fully unsupervised feature selection methods not designed for the CA setting. An open-source implementation of our method is available at `https://github.com/suinleelab/CFS`.

## 1 Introduction

In many scientific domains, it is increasingly common to analyze high-dimensional datasets consisting of observations with many features [1, 2, 3]. However, not all features are created equal: noisy, information-poor features can obscure important structures in a dataset and impair the performance of machine learning algorithms on downstream tasks. As such, it is often desired to select a small number of informative features to consider in subsequent analyses. For example, biologists may seek to measure the expression levels of a small number of genes and use these in a variety of future prediction tasks, such as disease subtype prediction or cell type classification. Measuring only a subset of features may also yield other benefits, including reduced experimental costs, enhanced interpretability, and better generalization for models trained with limited data.

While feature selection is relatively straightforward when supervision guides the process (e.g., class labels), such information is often unavailable in practice. In the unsupervised setting, we can select features for specific objectives like preserving clusters [4, 5] or local structure in the data [6, 7], or we can aim to simply reconstruct the full feature vector [8, 3]. However, these approaches are likely to focus on factors that dominate in the dataset, even if these structures are uninteresting for a specific

37th Conference on Neural Information Processing Systems (NeurIPS 2023).

downstream analysis. A third setting that lies between supervised and unsupervised is *contrastive analysis* [9, 10, 11], which we describe below.

In the contrastive analysis (CA) setting, we have two datasets conventionally referred to as the *target* and *background* [10]. The target dataset contains observations with variability from both "interesting" (e.g., due to a specific disease under study) and "uninteresting" (e.g., demographic-related) sources. On the other hand, the corresponding background dataset (e.g., measurements from healthy patients with similar demographic profiles) is assumed to only contain variations due to the uninteresting sources. The goal of CA is then to isolate the patterns enriched in the target dataset and not present in the background for further study. Here, the target-background dataset pairing provides a form of weak supervision to guide our analysis, and carefully exploiting their differences can help us focus on features that are maximally informative of the target-specific variations.

For example, suppose our goal were to measure gene expression in cancer patients for a subset of genes to better understand molecular subtypes of cancer. Identifying factors specific to cancerous tissues is of great importance to the cancer research community, as these intra-cancer variations are essential to study disease progression and treatment response [12]. However, selecting an appropriate set of genes is not straightforward, as unsupervised methods are likely to be confounded by variations due to nuisance factors such as age, ethnicity, or batch effects, which often dominate the overall variation in gene expression and other omics datasets (genomics, transcriptomics, proteomics) [10, 13, 14]. Moreover, fully supervised methods may not be applicable as they require detailed labels that may not be available *a priori* (e.g., due to the rapid rate of development and testing of new cancer treatments [15]). Thus, methods that can effectively leverage the weak supervision available in CA may be able to select more appropriate features for a given analysis.

Isolating salient variations is the focus of many recent works on CA, including many that generalize unsupervised representation learning methods to the CA setting [9, 10, 11, 16, 17, 18, 19, 13]. This work aims to develop a principled approach for *contrastive feature selection*, which represents an important use case not considered in prior work.[1]

The remainder of this manuscript is organized as follows: we begin (Section 2) with an introduction to the contrastive analysis setting and the unique challenges therein, followed by a discussion of related work (Section 3). We then proceed propose CFS, a feature selection method that leverages the weak supervision available in the CA setting (Section 4). CFS selects features that reflect variations enriched in a target dataset but are not present in the background data, and it does so using a novel two-step procedure that optimizes the feature subset within a neural network module. Subsequently, we develop a novel information-theoretic characterization of representation learning in the CA setting to justify our proposed method (Section 5); under mild assumptions about the data generating process, we prove that our two-step learning approach maximizes the mutual information with target-specific variations in the data. Finally, we validate our approach empirically through extensive experiments on a semi-synthetic dataset introduced in prior work as well as four real-world biomedical datasets (Section 6). We find that CFS consistently outperforms supervised and unsupervised feature selection algorithms, including state-of-the-art methods that it builds upon.

## 2   Problem formulation

We now describe the problem of feature selection in the CA setting. Our analysis focuses on an observed variable $x \in \mathbb{R}^d$ that we assume depends on two latent sources of variation: a salient variable $s$ that represents interesting variations (e.g., disease subtype or progression), and a separate background variable $z$ that represents variations that are not relevant for the intended analysis (e.g., gender or demographic information). Thus, we assume that observations are drawn from the data distribution $p(x) = \int p(x \mid z, s)p(z, s)dzds$, which we depict using a graphical model in Figure 1.

Our goal is to select a small number of features that represent the relevant variability in the data, or that contain as much information as possible about $s$. To represent this, we let $S \subseteq [d] \equiv \{1, \ldots, d\}$ denote a subset of indices and $x_S \equiv \{x_i : i \in S\}$ a subset of features. For a given value $k < d$, we aim to find $S$ with $|S| = k$ where $x_S$ is roughly equivalent to $s$, or is similarly useful for downstream

---

[1]We note that contrastive analysis (CA) is **unrelated** to contrastive learning in the sense of SimCLR [20], CLIP [21], or other self-supervised learning methods [22]. We retain the name used in prior work [9, 10, 16, 17].

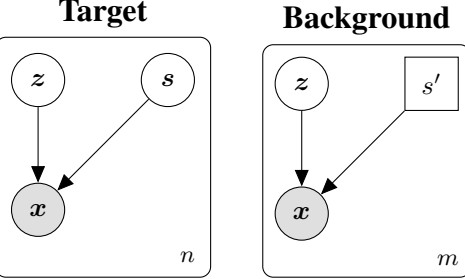

Figure 1: Data generating process for target and background samples. Observed values $x$ are generated from two latent variables: a salient variable $s$ and a background variable $z$. Both are active in the target dataset, while the salient variable is fixed at $s = s'$ in the background data.

tasks (e.g., clustering cells by disease status). We state this goal intuitively here following prior work [11], but we later frame our objective through the lens of mutual information (Section 5).

What distinguishes CA from the supervised setting is that we lack labels to guide the feature selection process [23]. We could instead perform unsupervised feature selection [3], but this biases our selections towards variations that dominate the data generating process; thus, in real-world genomics datasets, we may inadvertently focus on demographic features that are irrelevant to a specific disease. Instead, the CA setting provides a form of weak supervision via two separate unlabeled datasets [11]. First, we have a target dataset $\mathcal{D}_t$ consisting of samples with both sources of variation, or which follow the data distribution $p(x)$ defined above (Figure 1). Next, we have a background dataset $\mathcal{D}_b$ that consists of samples *with no salient variation*; this may represent a single fixed disease status, or perhaps no disease, and we formalize these as samples from $p(x \mid s = s')$ for a fixed value $s'$. We note that individual samples from the target and background datasets are not paired and we do not assume the same number of samples from the target and background datasets (i.e., $|\mathcal{D}_t| \neq |\mathcal{D}_b|$). By analyzing both datasets, we hope to distinguish sources of variability specific to $s$ rather than $z$.

The challenge we must consider is how to properly exploit the weak supervision provided by separate target and background datasets. To identify a maximally informative feature set $x_S$, we must effectively disentangle the distinct sources of variation in the data, and we wish to do so while allowing for different dataset sizes $|\mathcal{D}_b| = m$ and $|\mathcal{D}_t| = n$, greater variability in $x$ driven by irrelevant information $z$ than salient factors $s$, and with no need for additional labeling. Our work does so by building on previous work on contrastive representation learning [10, 16, 18] and neural feature selection [3, 23], while deriving insights that apply beyond the feature selection problem.

## 3  Related work

**Contrastive analysis**  Previous work has designed a number of representation learning methods to exploit the weak supervision available in the CA setting. Zou et al. [9] initially adapted mixture models for CA, and recent work adapted PCA to summarize target-specific variations while ignoring uninteresting factors of variability [10]. Other recent works developed probabilistic latent variable models [18, 16, 17, 19], including a method based on VAEs [11]. Severson et al. [16] experimented with adding a sparsity prior to perform feature selection within their model, but the penalty was designed for fully supervised scenarios: the penalty was manually tuned by observing how well pre-labeled target samples separated in the resulting latent space, which is not suitable for the scenario described in Section 2 where such labels are unavailable. Our work is thus the first to focus on feature selection in this setting, and we provide an information-theoretic analysis for our approach that generalizes to other forms of representation learning in the CA setting.

**Feature selection**  Feature selection has been extensively studied in the machine learning literature, see [24, 25] for recent reviews. Such methods can be divided broadly into filter [26, 27], wrapper [28, 29] and embedded [30, 31] methods, and they are also distinguished by whether they require labels (i.e., supervised versus unsupervised). Recent work has introduced mechanisms for neural feature selection [32, 3, 23, 33, 34, 35], which learn a feature subset jointly with an expressive model and allow a flexible choice of prediction target and loss function. Our work builds upon these methods, which represent the current state-of-the-art in feature selection. However, unlike prior works that

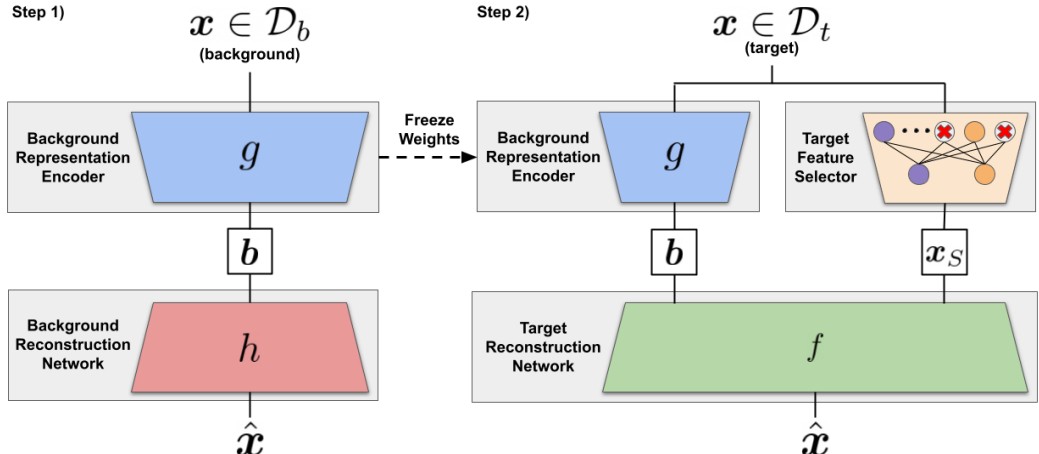

Figure 2: Overview of contrastive feature selection (CFS). Our approach consists of a two-step optimization procedure. In the first step (left), we train an autoencoder on a background dataset, which we assume is generated solely from irrelevant variations due to $z$. In the second step (right), we train a feature selector layer using our target dataset. To encourage the selection of features that best reflect salient variations due to $s$, the selector layer's output is concatenated with the output of the fixed encoder $g$, and then passed to a learned reconstruction function $f$. Ideally, the feature set $x_S$ chosen by the selector will capture only the factors of variation unique to the target data.

focus on either supervised [23, 34] or unsupervised feature selection [3, 36], we focus on contrastive dataset pairs and consider how to properly exploit the weak supervision in this setting.

## 4   Proposed method: contrastive feature selection

We aim to select $k$ features within $x \in \mathbb{R}^d$ that provide maximum information about the salient latent variable $s$ in our data generating process (Section 2). Such a procedure must be performed carefully, as standard unsupervised methods may optimize for selecting features that capture much of the overall variance in a dataset but not the (potentially subtle) variations of interest due to $s$.

To do so, we begin with the following intuitive idea: suppose we have *a priori* a variable $b \in \mathbb{R}^l$ with $l < d$ that serves as a proxy for $z$ (i.e., it summarizes all the uninteresting variation in the data). We assume for now that $b$ is already available, and that it should explain a significant portion of $x$'s variance, so that any remaining variance is due solely to factors of interest. We can then aim to find a complementary set of features that capture the remaining variations in $x$ not explained by $b$, and we do so by learning the feature set in conjunction with a function that reconstructs the full vector $x$. The reconstruction function $f: \mathbb{R}^{k+l} \mapsto \mathbb{R}^d$ uses both $b$ and the learned feature set $x_S$ as inputs, and we can learn $f$ and $S$ jointly as follows:

$$\underset{\theta, |S|=k}{\arg\min} \; \underset{x \sim \mathcal{D}_t}{\mathbb{E}} \; ||f(b, x_S; \theta) - x||^2. \tag{1}$$

We call this approach *contrastive feature selection* (CFS). Notably, eq. (1) requires data with both background and salient variations, and thus it uses only the target dataset $\mathcal{D}_t$. Intuitively, optimizing this objective will select the set of $k$ features $S$ that best capture the variations present in the target dataset but which are not captured by the variable $b$.

To implement CFS we must make two choices: (1) how to optimize the feature set $S \subseteq [d]$, and (2) how to obtain $b$. First, assuming we already have $b$, we can leverage recent techniques for differentiable feature selection to learn $S$ [23, 3]. Such methods select features by optimizing continuous approximations of discrete variables at the input layer of a neural network: selected variables are multiplied with a 1 and allowed to pass through the gates, while rejected features are multiplied with a 0. In our implementation of CFS, we use the Gaussian-based relaxation of Bernoulli

random variables termed *stochastic gates* (STG), proposed by Yamada et al. [23].[2] Let $\boldsymbol{G} \in [0, 1]^d$ denote a random vector of STG values parameterized by $\boldsymbol{\mu} \in \mathbb{R}^d$, so that samples of $\boldsymbol{G}$ are defined as

$$\boldsymbol{G}_i = \max(0, \min(1, \mu_i + \zeta)), \tag{2}$$

where $\mu_i$ is a learnable parameter and $\zeta$ is drawn from $\mathcal{N}(0, \sigma^2)$. By multiplying each feature by its corresponding gate value $\boldsymbol{G}_i$ and incorporating a regularization term that penalizes open gates (i.e., the number of indices with $\mu_i > 0$), feature selection can be performed in a fully differentiable manner. Leveraging the STG mechanism, we obtain the following continuous relaxation of eq. (1):

$$\underset{\boldsymbol{\mu}, \theta}{\arg\min} \; \underset{\boldsymbol{x} \sim \mathcal{D}_t}{\mathbb{E}} ||f(\boldsymbol{b}, \boldsymbol{x} \odot \boldsymbol{G}; \theta) - \boldsymbol{x}||^2 + \lambda \sum_{i=1}^{d} \Phi(\frac{\mu_i}{\sigma}), \tag{3}$$

where $\odot$ denotes the Hadamard product, $\lambda$ is a hyperparameter than controls the number of selected features, and $\Phi$ is the standard Gaussian CDF [23].

Next, we must decide how to obtain $\boldsymbol{b}$ so that it captures irrelevant variations due to $\boldsymbol{z}$, and ideally not those due to the salient variable $\boldsymbol{s}$. We propose to learn it as a function of $\boldsymbol{x}$, so that $\boldsymbol{b} = g(\boldsymbol{x}; \phi)$ and $g : \mathbb{R}^d \mapsto \mathbb{R}^l$. Intuitively, this is where we can leverage the background dataset, because it contains only uninteresting variations due to $\boldsymbol{z}$. We learn this background representation function in conjunction with a reconstruction network $h : \mathbb{R}^l \mapsto \mathbb{R}^d$ using the following objective:

$$\underset{\phi, \eta}{\arg\min} \; \underset{\boldsymbol{x} \sim \mathcal{D}_b}{\mathbb{E}} ||h(g(\boldsymbol{x}; \phi); \eta) - \boldsymbol{x}||^2. \tag{4}$$

Previous representation learning methods for CA, such as the contrastive VAEs of Abid and Zou [11] and Severson et al. [16], also considered learning two representations; however, they learned both representations *jointly* by training simultaneously with target and background data. In our feature selection context, this joint training setup would entail optimizing with the sum of eq. (3) and eq. (4), so that $g(\boldsymbol{x}; \phi)$ is updated based on both $\mathcal{D}_b$ and $\mathcal{D}_t$. This is potentially problematic, because the background representation $\boldsymbol{b} = g(\boldsymbol{x}; \phi)$ may absorb salient information that then cannot be learned by $\boldsymbol{x}_S$. Indeed, subsequent work observed that such joint training procedures may fail to properly segregate shared and target-specific variations, with target-specific variations being captured by the background representation function and vice versa [37].

In order to avoid this undesirable "leakage" of salient information into our background representation, we thus separate our objectives into a two-stage optimization procedure. For a given target-background dataset pair, we use the first stage to train an autoencoder network using *only background samples* (eq. (4)). In the second stage, we then use the encoder network $\boldsymbol{b} = g(\boldsymbol{x}; \phi)$ as our background proxy, freeze its weights, and learn a feature set $\boldsymbol{x}_S$ by optimizing eq. (3) using *only target samples*. We depict this procedure graphically in Figure 2. In the following section, we further motivate this two-stage approach by developing an information-theoretic formulation of representation learning in the CA setting, which illustrates why this approach compares favorably to joint contrastive training [11, 16], as well as fully unsupervised feature selection approaches [3].

## 5 Information-theoretic analysis

In this section, we motivate our approach from Section 4 through a framework of mutual information maximization. To justify learning two representations that are optimized in separate steps, we frame each step as maximizing a specific information-theoretic objective, which we then relate to our ultimate goal of representing the salient variation $\boldsymbol{s}$ via a feature set $\boldsymbol{x}_S$.

We simplify our analysis by considering that all variables $(\boldsymbol{x}, \boldsymbol{s}, \boldsymbol{z})$ are all discrete rather than continuous, which guarantees that they have non-negative entropy [38]. The learned representations are deterministic functions of $\boldsymbol{x}$, and we generalize our analysis by letting the representation from the second step be an arbitrary variable $\boldsymbol{a}$ rather than a feature set $\boldsymbol{x}_S$. Thus, our analysis applies even beyond the feature selection setting.

---

[2]While we focused on the STG layer in this work, CFS can be implemented with other choices of selection layers (e.g. the concrete selector layer of [3]). We refer the reader to Appendix A for further details.

## 5.1 Analyzing our approach

Before presenting our main results, we state two sets of assumptions required for our analysis. Regarding the data generating process (see Figure 1), we require mild conditions about the latent variables being independent and about each variable being explained by the others.

**Assumption 1.** *(Assumptions on data distribution.) We assume that the latent variables $s, z$ are roughly independent, that $x$ is well-explained by the latent variables, and that each latent variable is well-explained by $x$ and the other latent variable. That is, we assume a small constant $\epsilon > 0$ such that $I(s; z) \leq \epsilon$, $H(x \mid s, z) \leq \epsilon$, $H(s \mid x, z) \leq \epsilon$, and $H(z \mid x, s) \leq \epsilon$.*

Intuitively, we require independence so that salient information is not learned in the first step with $b$ and then ignored by $a$; the other conditions guarantee that $s$ can be recovered at all via $x$. We also require an assumption about the learned representations, which is that they do not affect the independence between the latent variables $s, z$. This condition is not explicitly guaranteed by our approach, but it also is not discouraged, so we view it as a mild assumption.

**Assumption 2.** *(Assumptions on learned representations.) We assume that $s, z$ remain roughly independent after conditioning on the learned representations $a, b$. That is, we assume a small constant $\epsilon > 0$ such that $I(s; z \mid b) \leq \epsilon$ and $I(s; z \mid a) \leq \epsilon$.*

With these assumptions, we can now present our analysis. First, we re-frame each step of our approach in an information-theoretic fashion:

- The first step in Section 4 is analogous to maximizing the mutual information $I(b; x \mid s = s')$. We do this for only a single value $s'$ in practice, but for our analysis we consider that this is similar to $I(b; x \mid s)$, which would be achieved if we optimized $b$ over multiple background datasets with different values. This holds because $I(b; x \mid s) = \mathbb{E}_s[I(b; x \mid s = s)]$.

- The second step in Section 4 is analogous to maximizing the mutual information $I(a, b; x)$. However, by treating $b$ as fixed and rewriting the objective as $I(a, b; x) = I(a; x \mid b) + I(b; x)$, we see that we are in fact only maximizing $I(a; x \mid b)$, or encouraging $a$ to complement $b$.

- Finally, our true objective is for $a$ to be equivalent to $s$, which we view as maximizing $I(a; s)$. If their mutual information is large, then $a$ is as useful as $s$ in downstream prediction tasks [38]. The challenge in CA is optimizing this when we lack labeled training data.

Now, we have the following result that shows our two-step procedure implicitly maximizes $I(a; s)$, which is our true objective in contrastive analysis (proof in Appendix B).

**Theorem 1.** *Under Assumptions 1 and 2, learning $b$ by maximizing $I(b; x \mid s)$ and learning $a$ by maximizing $I(a; x \mid b)$ yields the following lower bound on the mutual information $I(a; s)$:*

$$I(a; x \mid b) + I(b; x \mid s) - H(z) - 4\epsilon \leq I(a; s). \tag{5}$$

The result above shows that our true objective is related to $I(b; x \mid s)$ and $I(a; x \mid b)$, which represent the two steps of our learning approach. Specifically, the sum of these terms provides a lower bound on $I(a; s)$, so maximizing them implicitly maximizes $I(a; s)$ (similar to the ELBO in variational inference [39]). Within the inequality, $H(z)$ acts as a rough upper limit on $I(b; x \mid s)$, and the constant $\epsilon$ from Assumptions 1 and 2 contributes to looseness in the bound, which becomes tight in the case where $\epsilon = 0$. This result requires several steps to show, but it is perhaps intuitive that letting $b$ serve as a proxy for $z$ and then learning $a$ as a complementary variable helps recover the remaining variation, or encourages a large value for $I(a; s)$.

## 5.2 Comparison to other approaches

We now compare our result for our two-stage approach with two alternatives: (1) a simpler one-stage "joint" training approach similar to previously proposed contrastive VAEs [11, 16], and (2) an unsupervised learning approach similar to the concrete autoencoder of Balın et al. [3].

First, the joint training approach learns $a$ and $b$ simultaneously rather than separately, which we can view as optimizing one objective corresponding to each dataset (background and target). Following the argument in Section 5.1, the background data is used roughly to maximize $I(b; x \mid s)$. However,

the target data is used to maximize $I(\boldsymbol{a}, \boldsymbol{b}; \boldsymbol{x})$ with respect to both $\boldsymbol{a}$ and $\boldsymbol{b}$ rather than just $\boldsymbol{a}$. Thus, we can modify the inequality in eq. (5) to obtain the following lower bound on $I(\boldsymbol{a}; \boldsymbol{s})$:

$$I(\boldsymbol{a}, \boldsymbol{b}; \boldsymbol{x}) + I(\boldsymbol{b}; \boldsymbol{x} \mid \boldsymbol{s}) - \underbrace{I(\boldsymbol{b}; \boldsymbol{x})}_{\substack{\text{Maximized} \\ \text{during training}}} - H(\boldsymbol{z}) - 4\epsilon \leq I(\boldsymbol{a}; \boldsymbol{s}). \tag{6}$$

This follows directly from Theorem 1 and the definition of mutual information [38]. By maximizing $I(\boldsymbol{a}, \boldsymbol{b}; \boldsymbol{x}) = I(\boldsymbol{b}; \boldsymbol{x}) + I(\boldsymbol{a}; \boldsymbol{x} \mid \boldsymbol{b})$ rather than just $I(\boldsymbol{a}; \boldsymbol{x} \mid \boldsymbol{b})$, we may inadvertently reduce the lower bound through $I(\boldsymbol{b}; \boldsymbol{x})$ and therefore limit the salient information that $\boldsymbol{a}$ is forced to recover. Indeed, our experiments show that the joint training approach results in lower-quality features compared to our proposed two-stage method (Section 6).

Next, we consider fully unsupervised methods that learn a single representation $\boldsymbol{a}$. Such methods do not leverage paired contrastive datasets, so we formalize them as maximizing the mutual information $I(\boldsymbol{a}; \boldsymbol{x})$, which is natural when $\boldsymbol{a}$ is used to reconstruct $\boldsymbol{x}$ [3, 40]. Intuitively, if most variability in the data is due to irrelevant variation in $\boldsymbol{z}$, these methods are encouraged to maximize $I(\boldsymbol{a}; \boldsymbol{z})$ rather than $I(\boldsymbol{a}; \boldsymbol{s})$. This can be seen through the following result (see proof in Appendix B).

**Theorem 2.** *When we learn a single representation $\boldsymbol{a}$ by maximizing $I(\boldsymbol{a}; \boldsymbol{x})$, we obtain the following simultaneous lower bounds on $I(\boldsymbol{a}; \boldsymbol{s})$ and $I(\boldsymbol{a}; \boldsymbol{z})$:*

$$I(\boldsymbol{a}; \boldsymbol{x}) - H(\boldsymbol{x}) + I(\boldsymbol{x}; \boldsymbol{s}) \leq I(\boldsymbol{a}; \boldsymbol{s})$$
$$I(\boldsymbol{a}; \boldsymbol{x}) - H(\boldsymbol{x}) + I(\boldsymbol{x}; \boldsymbol{z}) \leq I(\boldsymbol{a}; \boldsymbol{z}). \tag{7}$$

The implication of Theorem 2 is that if $\boldsymbol{z}$ explains more variation than $\boldsymbol{s}$ in the observed variable $\boldsymbol{x}$, or if we have $I(\boldsymbol{x}; \boldsymbol{z}) > I(\boldsymbol{x}; \boldsymbol{s})$, then we obtain a higher lower bound for $I(\boldsymbol{a}; \boldsymbol{z})$ than $I(\boldsymbol{a}; \boldsymbol{s})$. This is undesirable, because the mutual information with $\boldsymbol{x}$ roughly decomposes between $\boldsymbol{s}$ and $\boldsymbol{z}$: as we show in Appendix B, Assumptions 1 and 2 imply that we have $|I(\boldsymbol{a}; \boldsymbol{s}) + I(\boldsymbol{a}; \boldsymbol{z}) - I(\boldsymbol{a}; \boldsymbol{x})| \leq 2\epsilon$.

We conclude that under this information-theoretic perspective and several mild assumptions, we can identify concrete benefits to a two-stage approach that leverages paired contrastive datasets.

### 5.3 Caveats and practical considerations

There are several practical points to discuss about our analysis, but we first emphasize the aim of this section: our goal is to justify a two-stage approach that leverages paired contrastive datasets, and to motivate performing the optimization in these steps separately rather than jointly. Our approach provides clear practical benefits (Section 6), and this analysis provides a lens to understand why that is, even if it does not perfectly describe how the algorithm works in practice.

The caveats to discuss are the following. (1) In practice, we minimize mean squared error for continuous variables rather than maximizing mutual information; however, the two are related, and Appendix C shows that minimizing mean squared error maximizes a lower bound on the mutual information. (2) Our analysis relates to an arbitrary variable $\boldsymbol{a}$ rather than a feature set $\boldsymbol{x}_S$. This is not an issue, and in fact implies that our results apply to more general representation learning and could improve existing methods [11]. (3) The assumptions we outlined may not hold in practice (see Assumptions 1 and 2). However, this represents a legitimate failure mode rather than an issue in our analysis: for example, if the latent variables are highly dependent, the second step may not recover meaningful variation from $\boldsymbol{s}$ because this will be captured to some extent by $\boldsymbol{b}$ in the first step.

## 6 Experiments

Here, we empirically validate our proposed CFS approach by using it to select maximally informative features in multiple target-background dataset pairs that were analyzed in previous work. We begin by considering a semi-synthetic dataset, Grassy MNIST [10], which lets us control the data generation process and the strength of uninteresting variations; we then report results on four publicly available biomedical datasets. For each target-background dataset pair, samples in the target dataset have ground-truth class labels related to the salient variations, which we use to evaluate the selected features. However, distinguishing these classes may be difficult using methods not designed for CA.

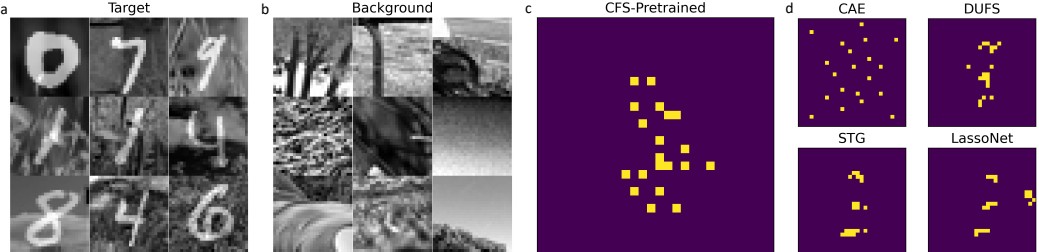

Figure 3: **a-b**, Example target (**a**) and background (**b**) images from the Grassy MNIST dataset. Our goal here is to select the set of features that best capture the target-dataset-specific variations due to the MNIST digits. **c-d**, $k = 20$ features selected by CFS (**c**) versus unsupervised (CAE [3], DUFS [36]) and supervised (STG [23], LassoNet [34]) baselines (**d**).

To illustrate the benefits of leveraging the weak supervision available in the CA setting, we compared against unsupervised feature selection methods applied to our target datasets. In particular, we considered two recently proposed methods that claim state-of-the-art results in fully unsupervised settings: the concrete autoencoder (CAE) [3] and differentiable unsupervised feature selection (DUFS) [36]. Moreover, to illustrate that naive adaptations of previous methods designed for fully supervised settings are not sufficient to fully exploit the weak supervision available in the CA setting, we compared against the state-of-the-art fully supervised methods STG [23] and LassoNet [34] trained to simply classify samples as target versus background.

Finally, to illustrate the importance of carefully learning CFS's background representation function $g$ (as discussed in Sections 4 and 5), we experimented with three variations of CFS. First, we performed the two-stage approach proposed in Section 4, where $g$ is pretrained on background data (CFS-Pretrained). Second, we tried learning $g$ jointly with our feature set (CFS-Joint), similar to previous work on representation learning for CA [11, 16]. Finally, in an attempt to combine the benefits of our proposed approach (i.e., improved separation of salient and irrelevant variations) with the simplicity of joint training (i.e., the ability to carry out optimization in a single step), we experimented with a modification of the joint training procedure: for target data points, we applied a stop-gradient operator to the output of the background representation function $g$ (CFS-SG). By doing so, $g$ is only updated for background data points and thus is not encouraged to pick up target-specific variations.

We refer the reader to Appendix D for details on model implementation and training.[3] For all experiments we divided our data using an 80-20 train-test split, and we report the mean $\pm$ standard error over five random seeds for each method.

### 6.1 Semi-synthetic data: Grassy MNIST

The Grassy MNIST dataset, originally introduced in Abid et al. [10], is constructed by superimposing pictures with the "grass" label from ImageNet [41] onto handwritten digits from the MNIST dataset [42]. For each MNIST digit, a randomly chosen grass image was cropped, resized to be 28 by 28 pixels, and scaled to have double the amplitude of the MNIST digit before being superimposed onto the digit (Figure 3a). For a background dataset we used a separate set of grass images with no digits (Figure 3b), and our goal here is to select features that best capture the target-specific digit variations.

We began by using our CFS variants with a fixed background dimension $l = 20$ and baseline models to select $k = 20$ features. Qualitatively, the features selected by CFS (Figure 3c; Figure S1a-b) concentrate around the center where digits are found, as desired. On the other hand, features selected by baselines (Figure 3d) are undesirably located far from the center where digits are never found (CAE), or are clustered in only a subset of locations where digit-related variations can be found (STG, DUFS), or both (LassoNet). These results illustrate the unsuitability of fully unsupervised methods for CA, as such methods may select features that contain minimal information on the target-specific salient variations. Moreover, these results illustrate that naive adaptations of supervised methods are not suitable for CA, as features that are sufficient to discriminate between target and background samples do not necessarily contain maximal information on the target-specific salient variations.

---

[3]Code for reproducing our results is available in supplementary materials.

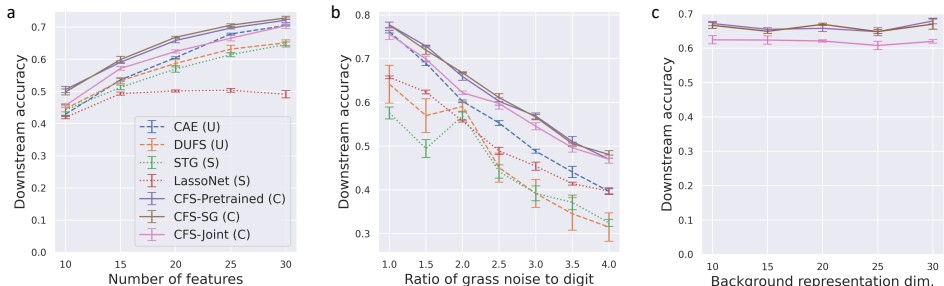

Figure 4: **a**, Quantitative performance comparison of CFS versus baseline supervised (S) and unsupervised (U) methods on Grassy MNIST for varying numbers of features $k$. **b**, We vary the relative contribution of grass noise to the dataset and assess each method's performance when used to select $k = 20$ features. **c**, We select $k = 20$ features on the original Grassy MNIST dataset and vary CFS's background representation dimension $l$ to understand the impact of this hyperparameter.

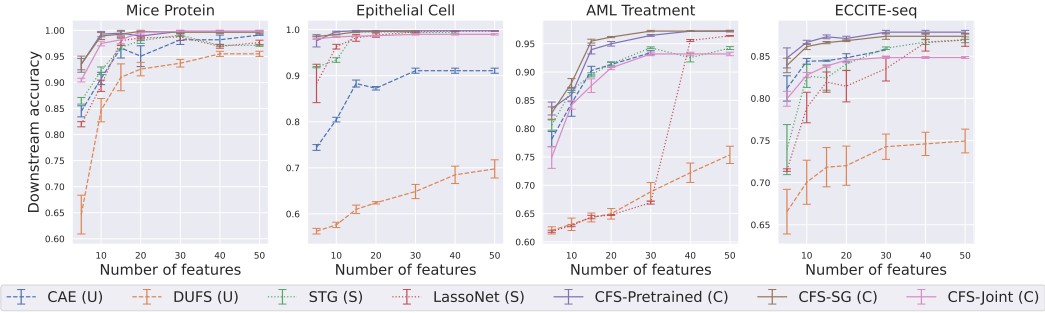

Figure 5: Benchmarking CFS and baseline supervised (S) and unsupervised (U) methods on four real-world biomedical datasets.

We then evaluated the quality of features selected by each method by training random forest classifiers to predict the digit (i.e., 0-9) in each image, using only the $k$ selected features. We report the accuracy of the classifiers on a held out test set in Figure 4a for varying values of $k$. We find that our pretrained and stop-gradient CFS models outperform baseline methods, with a minimal gap between them. On the other hand, CFS-Joint selected lower quality features and did not always outperform baselines, reflecting the pitfalls of joint training discussed in Sections 4-5. We next assessed how CFS's performance varies with respect to the ratio of background noise vs. salient variation by varying the scale of grass noise versus MNIST digit: when selecting $k = 20$ features at each increment, we found that CFS continued to outperform baselines across noise scales (Figure 4b). Finally, we assessed the impact of the background representation size $l$ for our CFS variants by selecting $k = 20$ features on the original Grassy MNIST dataset with varying values of $l$, and we found that our results were largely insensitive to this hyperparameter (Figure 4c). We also found that this phenomenon was consistent across varying choices of $k$ (Figure S3).

## 6.2 Real-world datasets

We next applied CFS and baseline methods to four real-world biomedical datasets: protein expression levels from trisomic and control mice that were exposed to different experimental conditions (Mice Protein) [43], single-cell gene expression measurements from mice intestinal epithelial cells exposed to different pathogens (Epithelial Cell) [44], single-cell gene expression measurements from patients with acute myeloid leukemia before and after a bone marrow transplant (AML Treatment) [45], and gene expression measurements from a high-throughput single-cell gene knockout perturbation screen (ECCITE-seq) [14]. For each target dataset, we obtained a corresponding background collected as part of the same study. We refer readers to Appendix E for further details on each dataset.

Feature quality was again assessed by training random forest models to classify target samples into known subgroups of target points, and we report classification accuracy on a held-out test set for varying numbers of features in Figure 5. Once again we find that our pretrained and stop-gradient

CFS variants outperformed baseline fully supervised and unsupervised methods, demonstrating that carefully exploiting the weak supervision available in the CA setting can lead to higher-quality features for downstream CA tasks. Furthermore, we find that CFS-Joint's performance was inconsistent across datasets, and that it sometimes underperformed baseline models. This phenomenon further illustrates the need to carefully train the background function $g$ as discussed in Sections 4-5 to ensure the selection of features that are maximally informative for CA. Moreover, we found that these results were consistent for other choices of downstream classifier model, including XGBoost [46] and multilayer perceptrons (Figure S4). Using the mice protein dataset, we also performed further experiments assessing the importance of the background dataset for CFS's performance. In particular, we assessed the performance of our pretrained and gradient stopping CFS variants as the number of background samples varied, and we also compared against a variant of CFS where the background encoder was fixed at a random initialization (i.e., the background encoder was not updated during training). We found that CFS's performance initially improved as the number of background samples increased before saturating (Table S2), and we also found that the randomly initialized CFS variant exhibited significantly worse performance for lower numbers of features (Table S3).

Finally, to further illustrate the utility of our method for exploring real-world biomedical datasets, we performed a deeper analysis of the specific genes selected by CFS on the Epithelial Cell pathogen infection dataset, and we present these results in Appendix F. In short, we found that CFS tended to select genes that exhibited subtle cell-type-specific responses to pathogen infection. On the other hand, features selected by baseline methods were sufficient for distinguishing between control and infected cells, but did not capture these more subtle cell-type-specific effects. This result thus illustrates CFS' potential to facilitate new biological insights.

## 7  Discussion

In this work we considered the problem of feature selection in the contrastive analysis (CA) setting. Our goal is to discover features that reflect salient variations enriched in a target dataset compared to a background dataset, and we tackled the problem by proposing CFS (Section 4), a method we motivated with a new information-theoretic analysis of representation learning in the CA setting (Section 5). Experiments on both simulated and real-world datasets show that features selected by CFS led to superior performance on downstream CA tasks as compared to those selected by previous feature selection methods. CFS represents a promising method to identify informative features when salient variations are subtly expressed in unlabeled contrastive datasets, and our theoretical analysis may prove useful for developing and comparing other learning methods designed for the CA setting.

## Acknowledgements

This work was funded by NSF DBI-1552309 and DBI-1759487 (E.W., I.C. and S.-I.L.), NIH R35-GM-128638 and R01-NIA-AG-061132 (E.W., I.C. and S.-I.L.). E.W. was supported by the National Science Foundation Graduate Research Fellowship under Grant No. DGE-2140004.

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

# A    Choice of the feature selection layer

For the experiments presented in the main text, we implemented CFS with the STG feature selection layer proposed in Yamada et al. [23]. However, CFS does not have to be implemented with STG, and can be implemented with any choice of differentiable feature selection layer. Here we describe in more detail another potential choice of feature selection layer: the concrete selector layer introduced in Balın et al. [3].

The concrete selector layer is based on concrete random variables [47, 48]. A concrete random variable can be sampled to produce a continuous approximation of a one-hot vector. To sample a $d$-dimensional concrete random variable, one first samples a $d$-dimensional vector of i.i.d. samples from a Gumbel distribution [49] $\boldsymbol{g}$. Each element $\boldsymbol{m}$ of our concrete variable is then defined as

$$\boldsymbol{m}_i = \frac{\exp((\log \boldsymbol{\alpha}_i + \boldsymbol{g}_i)/T)}{\sum_{k=1}^{d} \exp((\log \boldsymbol{\alpha}_k + \boldsymbol{g}_k)/T)}$$

Here our temperature parameter $T \in (0, \infty)$ controls the extent to which our one-hot vector is relaxed. In the limit $T \to 0$, our concrete random variable outputs one-hot vectors with $\boldsymbol{m}_i = 1$ with probability $\boldsymbol{\alpha}_i / \sum_{k=1}^{d} \boldsymbol{\alpha}_k$. Concrete random variables are differentiable with respect to their parameters $\boldsymbol{\alpha}$ via the reparameterization trick [39].

The concrete selector layer then selects features in the following manner. For each of the $k$ nodes in the selector layer, we sample a $d$-dimensional concrete random variable $\boldsymbol{m}^{(i)}$, $i \in \{1, \ldots, k\}$. The $i$th node in the selector layer $u^{(i)}$ outputs $\boldsymbol{x} \cdot \boldsymbol{m}^{(i)}$. As $T \to 0$ this inner product simply becomes equal to one of the input features. After the network is trained, the concrete selector layer is replaced by an $\arg\max$ layer for which the output of the $i$th neuron is $\boldsymbol{x}_{\arg\max_j \boldsymbol{\alpha}_j}$.

Before training, the selector's parameters $\boldsymbol{\alpha}_j$ are initialized as small positive values to encourage the layer to explore different combinations of input features. As the layer is trained, the values of $\boldsymbol{\alpha}_j$ become more sparse as the layer becomes more confident in particular choices of features.

Similar to [3], we can train concrete selector layers with CFS using a simple annealing schedule for the temperature parameter $T$. That is, we begin training with a high temperature $T_0$ and gradually decay the temperature until we reach a final temperature $T_B$ at each epoch according to a first-order exponential decay $T(b) = T_0(T_B/T_0)^{b/B}$ where $T(b)$ is the temperature at epoch $b$ and $B$ is the total number of epochs.

In Figure S2 we present the results of applying CFS with the concrete selector layer to select features on the Grassy MNIST dataset. As with our original CFS implementation with the STG layer, we found that the pretrained and gradient-stopping CFS variants implemented with the concrete layer selected features around the center of images as desired.

# B Proofs

In this section, we re-state and then prove our theoretical results from the main text.

**Theorem 1.** *Under Assumptions 1 and 2, learning $\boldsymbol{b}$ by maximizing $I(\boldsymbol{b}; \boldsymbol{x} \mid \boldsymbol{s})$ and learning $\boldsymbol{a}$ by maximizing $I(\boldsymbol{a}; \boldsymbol{x} \mid \boldsymbol{b})$ yields the following lower bound on the mutual information $I(\boldsymbol{a}; \boldsymbol{s})$:*

$$I(\boldsymbol{a}; \boldsymbol{x} \mid \boldsymbol{b}) + I(\boldsymbol{b}; \boldsymbol{x} \mid \boldsymbol{s}) - H(\boldsymbol{z}) - 4\epsilon \leq I(\boldsymbol{a}; \boldsymbol{s}).$$

*Proof.* In the first step of our learning algorithm, we learn the representation $\boldsymbol{b}$ by maximizing $I(\boldsymbol{b}; \boldsymbol{x} \mid \boldsymbol{s})$. To show how this relates to maximizing $I(\boldsymbol{b}; \boldsymbol{z})$, consider the following joint mutual information, which we can decompose in two ways using the chain rule of mutual information [38]:

$$
\begin{aligned}
I(\boldsymbol{b}; \boldsymbol{s}, \boldsymbol{z}, \boldsymbol{x}) &= I(\boldsymbol{b}; \boldsymbol{z}) + I(\boldsymbol{b}; \boldsymbol{s} \mid \boldsymbol{z}) + I(\boldsymbol{b}; \boldsymbol{x} \mid \boldsymbol{s}, \boldsymbol{z}) \\
&= I(\boldsymbol{b}; \boldsymbol{s}) + I(\boldsymbol{b}; \boldsymbol{x} \mid \boldsymbol{s}) + I(\boldsymbol{b}; \boldsymbol{z} \mid \boldsymbol{x}, \boldsymbol{s}).
\end{aligned}
$$

Rearranging terms, we can see the following relationship between $I(\boldsymbol{b}; \boldsymbol{x} \mid \boldsymbol{s})$ and $I(\boldsymbol{b}; \boldsymbol{z})$:

$$I(\boldsymbol{b}; \boldsymbol{x} \mid \boldsymbol{s}) = I(\boldsymbol{b}; \boldsymbol{z}) + \underbrace{I(\boldsymbol{b}; \boldsymbol{x} \mid \boldsymbol{s}, \boldsymbol{z})}_{\delta_1} + \left( \underbrace{I(\boldsymbol{b}; \boldsymbol{s} \mid \boldsymbol{z}) - I(\boldsymbol{b}; \boldsymbol{s})}_{\delta_2} \right) - \underbrace{I(\boldsymbol{b}; \boldsymbol{z} \mid \boldsymbol{x}, \boldsymbol{s})}_{\delta_3}.$$

We can show that each of the $\delta_i$ terms is bounded near zero due to our assumptions:

$$
\begin{aligned}
\delta_1 &= I(\boldsymbol{b}; \boldsymbol{x} \mid \boldsymbol{s}, \boldsymbol{z}) \leq H(\boldsymbol{x} \mid \boldsymbol{s}, \boldsymbol{z}) \leq \epsilon && \Rightarrow \delta_1 \in [0, \epsilon] \\
\delta_2 &= I(\boldsymbol{b}; \boldsymbol{s} \mid \boldsymbol{z}) - I(\boldsymbol{b}; \boldsymbol{s}) = I(\boldsymbol{s}; \boldsymbol{z} \mid \boldsymbol{b}) - I(\boldsymbol{s}; \boldsymbol{z}) && \Rightarrow \delta_2 \in [-\epsilon, \epsilon] \\
\delta_3 &= I(\boldsymbol{b}; \boldsymbol{z} \mid \boldsymbol{x}, \boldsymbol{s}) \leq H(\boldsymbol{z} \mid \boldsymbol{x}, \boldsymbol{s}) \leq \epsilon && \Rightarrow \delta_3 \in [0, \epsilon]
\end{aligned}
$$

Thus, our objective in the first step has the property that $I(\boldsymbol{b}; \boldsymbol{x} \mid \boldsymbol{s}) \approx I(\boldsymbol{b}; \boldsymbol{z})$, or more precisely,

$$I(\boldsymbol{b}; \boldsymbol{x} \mid \boldsymbol{s}) - 2\epsilon \leq I(\boldsymbol{z}; \boldsymbol{b}) \leq I(\boldsymbol{b}; \boldsymbol{x} \mid \boldsymbol{s}) + 2\epsilon. \tag{8}$$

Regarding the second step of the learning algorithm, we can use a similar argument to show a relationship between $I(\boldsymbol{a}; \boldsymbol{x} \mid \boldsymbol{z})$ and $I(\boldsymbol{a}; \boldsymbol{s})$:

$$I(\boldsymbol{a}; \boldsymbol{x} \mid \boldsymbol{z}) = I(\boldsymbol{a}; \boldsymbol{s}) + \underbrace{I(\boldsymbol{a}; \boldsymbol{x} \mid \boldsymbol{s}, \boldsymbol{z})}_{\Delta_1} + \left( \underbrace{I(\boldsymbol{a}; \boldsymbol{z} \mid \boldsymbol{s}) - I(\boldsymbol{a}; \boldsymbol{z})}_{\Delta_2} \right) - \underbrace{I(\boldsymbol{a}; \boldsymbol{s} \mid \boldsymbol{x}, \boldsymbol{z})}_{\Delta_3}. \tag{9}$$

Because we lack access to $\boldsymbol{z}$ during this step, we must maximize $I(\boldsymbol{a}; \boldsymbol{x} \mid \boldsymbol{b})$ instead of $I(\boldsymbol{a}; \boldsymbol{x} \mid \boldsymbol{z})$. Fortunately, we can show that these are related using a similar mutual information decomposition:

$$
\begin{aligned}
I(\boldsymbol{a}; \boldsymbol{z}, \boldsymbol{x}, \boldsymbol{b}) &= I(\boldsymbol{a}; \boldsymbol{b}) + I(\boldsymbol{a}; \boldsymbol{x} \mid \boldsymbol{b}) + I(\boldsymbol{a}; \boldsymbol{z} \mid \boldsymbol{x}, \boldsymbol{b}) \\
&= I(\boldsymbol{a}; \boldsymbol{z}) + I(\boldsymbol{a}; \boldsymbol{x} \mid \boldsymbol{z}) + I(\boldsymbol{a}; \boldsymbol{b} \mid \boldsymbol{x}, \boldsymbol{z}).
\end{aligned}
$$

Rearranging terms, we get the following:

$$I(\boldsymbol{a}; \boldsymbol{x} \mid \boldsymbol{b}) = I(\boldsymbol{a}; \boldsymbol{x} \mid \boldsymbol{z}) + \underbrace{I(\boldsymbol{a}; \boldsymbol{b} \mid \boldsymbol{x}, \boldsymbol{z})}_{\Delta_4} + \left( \underbrace{I(\boldsymbol{a}; \boldsymbol{z}) - I(\boldsymbol{a}; \boldsymbol{b})}_{\Delta_5} \right) - \underbrace{I(\boldsymbol{a}; \boldsymbol{z} \mid \boldsymbol{x}, \boldsymbol{b})}_{\Delta_6}. \tag{10}$$

Putting the results from eq. (9) and eq. (10) together, we have the following:

$$I(\boldsymbol{a}; \boldsymbol{x} \mid \boldsymbol{b}) = I(\boldsymbol{a}; \boldsymbol{s}) + \Delta_1 + \Delta_2 - \Delta_3 + \Delta_4 + \Delta_5 - \Delta_6. \tag{11}$$

Combining our assumptions with the result from eq. (8), we can show that the $\Delta_i$ terms are bounded as follows:

$$\Delta_1 = I(\boldsymbol{a}; \boldsymbol{x} \mid \boldsymbol{s}, \boldsymbol{z}) \leq H(\boldsymbol{x} \mid \boldsymbol{s}, \boldsymbol{z}) \leq \epsilon \Rightarrow \Delta_1 \in [0, \epsilon]$$
$$\Delta_2 = I(\boldsymbol{a}; \boldsymbol{z} \mid \boldsymbol{s}) - I(\boldsymbol{a}; \boldsymbol{z}) = I(\boldsymbol{s}; \boldsymbol{z} \mid \boldsymbol{a}) - I(\boldsymbol{s}; \boldsymbol{z}) \Rightarrow \Delta_2 \in [-\epsilon, \epsilon]$$
$$\Delta_3 = I(\boldsymbol{a}; \boldsymbol{s} \mid \boldsymbol{x}, \boldsymbol{z}) \leq H(\boldsymbol{s} \mid \boldsymbol{x}, \boldsymbol{z}) \leq \epsilon \Rightarrow \Delta_3 \in [0, \epsilon]$$
$$\Delta_4 = I(\boldsymbol{a}; \boldsymbol{b} \mid \boldsymbol{x}, \boldsymbol{z}) \leq H(\boldsymbol{b} \mid \boldsymbol{x}) = 0 \Rightarrow \Delta_4 = 0$$
$$\Delta_5 = I(\boldsymbol{a}; \boldsymbol{z}) - I(\boldsymbol{a}; \boldsymbol{b}) = I(\boldsymbol{a}; \boldsymbol{z} \mid \boldsymbol{b}) - I(\boldsymbol{a}; \boldsymbol{b} \mid \boldsymbol{z}) \Rightarrow \Delta_5 \in [-H(\boldsymbol{b}) + I(\boldsymbol{b}; \boldsymbol{x} \mid \boldsymbol{s}) - 2\epsilon, H(\boldsymbol{z}) - I(\boldsymbol{b}; \boldsymbol{x} \mid \boldsymbol{s}) + 2\epsilon]$$
$$\Delta_6 = I(\boldsymbol{a}; \boldsymbol{z} \mid \boldsymbol{x}, \boldsymbol{b}) \leq H(\boldsymbol{z} \mid \boldsymbol{b}) = H(\boldsymbol{z}) - I(\boldsymbol{z}; \boldsymbol{b}) \Rightarrow \Delta_6 \in [0, H(\boldsymbol{z}) - I(\boldsymbol{b}; \boldsymbol{x} \mid \boldsymbol{s}) + 2\epsilon]$$

This yields the following two-sided inequality, which connects the empirical objective $I(\boldsymbol{a}; \boldsymbol{x} \mid \boldsymbol{b})$ to our implicit objective $I(\boldsymbol{a}; \boldsymbol{s})$:

$$I(\boldsymbol{a}; \boldsymbol{s}) + 2I(\boldsymbol{b}; \boldsymbol{x} \mid \boldsymbol{s}) - H(\boldsymbol{z}) - H(\boldsymbol{b}) - 6\epsilon \leq I(\boldsymbol{a}; \boldsymbol{x} \mid \boldsymbol{b}) \leq I(\boldsymbol{a}; \boldsymbol{s}) - I(\boldsymbol{b}; \boldsymbol{x} \mid \boldsymbol{s}) + H(\boldsymbol{z}) + 4\epsilon.$$

Focusing on a one-sided version of the result, we can see that the two empirical objectives $I(\boldsymbol{a}; \boldsymbol{x} \mid \boldsymbol{b})$ and $I(\boldsymbol{b}; \boldsymbol{x} \mid \boldsymbol{s})$ control the underlying objective $I(\boldsymbol{a}; \boldsymbol{s})$ as follows:

$$I(\boldsymbol{a}; \boldsymbol{x} \mid \boldsymbol{b}) + I(\boldsymbol{b}; \boldsymbol{x} \mid \boldsymbol{s}) - H(\boldsymbol{z}) - 4\epsilon \leq I(\boldsymbol{a}; \boldsymbol{s}).$$

$\square$

**Theorem 2.** *When we learn a single representation $\boldsymbol{a}$ by maximizing $I(\boldsymbol{a}; \boldsymbol{x})$, we obtain the following simultaneous lower bounds on $I(\boldsymbol{a}; \boldsymbol{s})$ and $I(\boldsymbol{a}; \boldsymbol{z})$:*
$$I(\boldsymbol{a}; \boldsymbol{x}) - H(\boldsymbol{x}) + I(\boldsymbol{x}; \boldsymbol{s}) \leq I(\boldsymbol{a}; \boldsymbol{s})$$
$$I(\boldsymbol{a}; \boldsymbol{x}) - H(\boldsymbol{x}) + I(\boldsymbol{x}; \boldsymbol{z}) \leq I(\boldsymbol{a}; \boldsymbol{z}).$$

*Proof.* Consider the following joint mutual information, which we can decompose in two ways:

$$\begin{aligned} I(\boldsymbol{a}; \boldsymbol{x}, \boldsymbol{s}) &= I(\boldsymbol{a}; \boldsymbol{s}) + I(\boldsymbol{a}; \boldsymbol{x} \mid \boldsymbol{s}) \\ &= I(\boldsymbol{a}; \boldsymbol{x}) + I(\boldsymbol{a}; \boldsymbol{s} \mid \boldsymbol{x}). \end{aligned}$$

Rearranging terms, we can see the following relationship between $I(\boldsymbol{a}; \boldsymbol{s})$ and $I(\boldsymbol{a}; \boldsymbol{x})$:

$$I(\boldsymbol{a}; \boldsymbol{s}) = I(\boldsymbol{a}; \boldsymbol{x}) + I(\boldsymbol{a}; \boldsymbol{s} \mid \boldsymbol{x}) - I(\boldsymbol{a}; \boldsymbol{x} \mid \boldsymbol{s}).$$

Bounding the remaining terms, we arrive at the following two-sided result:

$$I(\boldsymbol{a}; \boldsymbol{x}) - H(\boldsymbol{x} \mid \boldsymbol{s}) \leq I(\boldsymbol{a}; \boldsymbol{s}) \leq I(\boldsymbol{a}; \boldsymbol{x}) + H(\boldsymbol{s} \mid \boldsymbol{x}).$$

Our result in Theorem 2 focuses on the left-hand side and uses the definition of $I(\boldsymbol{x}; \boldsymbol{s})$. An identical argument can be used to show the same result for $I(\boldsymbol{a}; \boldsymbol{z})$.

$\square$

**Proposition 1.** *Under Assumptions 1 and 2, the representation $\boldsymbol{a}$ has mutual information with $\boldsymbol{x}$ that decomposes additively between $\boldsymbol{s}$ and $\boldsymbol{z}$:*

$$-\epsilon \leq I(\boldsymbol{a}; \boldsymbol{x}) - I(\boldsymbol{a}; \boldsymbol{s}) - I(\boldsymbol{a}; \boldsymbol{z}) \leq 2\epsilon.$$

*Proof.* Consider the mutual information $I(\boldsymbol{a}; \boldsymbol{x})$ for a representation $\boldsymbol{a}$ generated as a function of $\boldsymbol{x}$, which we can rewrite using the data processing inequality and chain rule:

$$\begin{aligned}
I(\boldsymbol{a}; \boldsymbol{x}) = I(\boldsymbol{a}; \boldsymbol{x}, \boldsymbol{s}, \boldsymbol{z}) &= I(\boldsymbol{a}; \boldsymbol{s}) + I(\boldsymbol{a}; \boldsymbol{z} \mid \boldsymbol{s}) + I(\boldsymbol{a}; \boldsymbol{x} \mid \boldsymbol{s}, \boldsymbol{z}) \\
&= I(\boldsymbol{a}; \boldsymbol{s}) + I(\boldsymbol{a}; \boldsymbol{z}) + (I(\boldsymbol{a}; \boldsymbol{z} \mid \boldsymbol{s}) - I(\boldsymbol{a}; \boldsymbol{z})) + I(\boldsymbol{a}; \boldsymbol{x} \mid \boldsymbol{s}, \boldsymbol{z}). \quad (12)
\end{aligned}$$

Assumption 1 implies that $0 \leq I(\boldsymbol{a}; \boldsymbol{x} \mid \boldsymbol{s}, \boldsymbol{z}) \leq H(\boldsymbol{x} \mid \boldsymbol{s}, \boldsymbol{z}) \leq \epsilon$ for the final term. For the subtraction term, we may expect that $I(\boldsymbol{a}; \boldsymbol{z} \mid \boldsymbol{s}) \approx I(\boldsymbol{a}; \boldsymbol{z})$ because $\boldsymbol{z}, \boldsymbol{s}$ are roughly independent. We can formalize this idea using another mutual information decomposition:

$$\begin{aligned}
I(\boldsymbol{a}, \boldsymbol{s}; \boldsymbol{z}) &= I(\boldsymbol{a}; \boldsymbol{z}) + I(\boldsymbol{s}; \boldsymbol{z} \mid \boldsymbol{a}) \\
&= I(\boldsymbol{s}; \boldsymbol{z}) + I(\boldsymbol{a}; \boldsymbol{z} \mid \boldsymbol{s}) \\
\Rightarrow I(\boldsymbol{a}; \boldsymbol{z} \mid \boldsymbol{s}) - I(\boldsymbol{a}; \boldsymbol{z}) &= I(\boldsymbol{s}; \boldsymbol{z} \mid \boldsymbol{a}) - I(\boldsymbol{s}; \boldsymbol{z}).
\end{aligned}$$

Assumptions 1 and 2 imply that $|I(\boldsymbol{a}; \boldsymbol{z} \mid \boldsymbol{s}) - I(\boldsymbol{a}; \boldsymbol{z})| \leq \epsilon$. Substituting this into eq. (12) yields our final result:

$$-\epsilon \leq I(\boldsymbol{a}; \boldsymbol{x}) - I(\boldsymbol{a}; \boldsymbol{s}) - I(\boldsymbol{a}; \boldsymbol{z}) \leq 2\epsilon.$$

$\square$

## C Connection between mean squared error and mutual information

When predicting $\boldsymbol{x}$ given an arbitrary representation $\boldsymbol{a}$, our loss function based on the reconstruction function $g$ is the following:

$$\min_g \mathbb{E}_{\boldsymbol{x}\boldsymbol{a}} \left[||g(\boldsymbol{a}) - \boldsymbol{x}||^2\right].$$

To minimize the loss, the best possible prediction is given by the function $g(a) = \mathbb{E}[\boldsymbol{x} \mid a]$, and our loss becomes the trace of the expected conditional variance:

$$\mathbb{E}_{\boldsymbol{x}\boldsymbol{a}} \left[|| \mathbb{E}[\boldsymbol{x} \mid \boldsymbol{a}] - \boldsymbol{x}||^2\right] = \text{Tr} \left(\mathbb{E}_{\boldsymbol{a}} \left[\text{Var}(\boldsymbol{x} \mid \boldsymbol{a})\right]\right).$$

Optimizing the mean squared error error with respect to $\boldsymbol{a}$ can thus be understood as minimizing the trace of $\boldsymbol{x}$'s expected conditional variance.

Next, we can show that the conditional variance is connected to the mutual information $I(\boldsymbol{a}; \boldsymbol{x})$. First, for a fixed value of $a$ where $\boldsymbol{x}$ has conditional variance $\text{Var}(\boldsymbol{x} \mid a)$, we have the following upper bound on the differential entropy,

$$H(\boldsymbol{x} \mid a) \leq \frac{1}{2} \log \det \left(\text{Var}(\boldsymbol{x} \mid a)\right) + \frac{d}{2} \log(2\pi e),$$

where $\det(\text{Var}(\boldsymbol{x} \mid a))$ denotes the determinant of the covariance matrix. This result follows from the fact that the entropy for random variables with fixed covariance is maximized by a Gaussian distribution [38]. Next, assuming that $\text{Var}(\boldsymbol{x} \mid a)$ is positive definite, or that $\text{Var}(\boldsymbol{x} \mid a) \succ 0$, we can modify the bound to use the trace rather than the log-determinant of the covariance matrix:

$$H(\boldsymbol{x} \mid a) \leq \frac{1}{2}\text{Tr}(\text{Var}(\boldsymbol{x} \mid a)) + \frac{d}{2} \log(2\pi).$$

Taking the expectation of both sides with respect to $\boldsymbol{a}$, we obtain the following relationship between the conditional entropy and expected conditional variance:

$$H(\boldsymbol{x} \mid \boldsymbol{a}) \leq \frac{1}{2}\text{Tr} \left(\mathbb{E}_{\boldsymbol{a}} \left[\text{Var}(\boldsymbol{x} \mid \boldsymbol{a})\right]\right) + \frac{d}{2} \log(2\pi).$$

Finally, using the relationship between the mutual information $I(\boldsymbol{a}; \boldsymbol{x})$ and conditional entropy $H(\boldsymbol{x} \mid \boldsymbol{a})$, we can identify the following connection between the minimum mean squared error and the mutual information achieved by $\boldsymbol{a}$:

$$H(\boldsymbol{x}) - \frac{d}{2} \log(2\pi) - \frac{1}{2} \left\{\min_g \mathbb{E}_{\boldsymbol{x}\boldsymbol{a}} \left[||g(\boldsymbol{a}) - \boldsymbol{x}||^2\right]\right\} \leq I(\boldsymbol{a}; \boldsymbol{x}).$$

Thus, minimizing the mean squared error with respect to $\boldsymbol{a}$ can be understood as maximizing a lower bound on the mutual information $I(\boldsymbol{a}; \boldsymbol{x})$. This shows that our empirical procedure, which involves minimizing mean squared error, is connected to our theoretical results that are stated in terms of mutual information.

We note that this bound can be vacuous if $H(\boldsymbol{x}) < 0$, which is possible for differential entropy, because we trivially have $I(\boldsymbol{a}; \boldsymbol{x}) \geq 0$ even for continuous random variables. The result should therefore not be interpreted as providing a precise estimate of the mutual information; it instead demonstrates the rough intuition that improving the mean squared error is related to increasing the mutual information.

## D   Implementation details

All experiments were peformed on a system running CentOS 7.9.2009 equipped with an NVIDIA RTX 2080 TI GPU with CUDA 11.7.

CFS models were implemented using PyTorch [50] (version 1.13) with the PyTorch Lightning API[4]. For all CFS variants we let our reconstruction function $f$ be a multilayer perceptron with two hidden layers of size 512 with ReLU activation functions. For our Joint and stop-gradient (SG) CFS models, we let $g$ be a multilayer perceptron with a single hidden layer of size 128. For our Pretrained CFS models, we first trained an autoencoder consisting of an encoder with a single hidden layer of size 128 and a decoder with the same architecture in reverse. The encoder network's weights were then frozen and it was used as $g$. All CFS models were trained using the PyTorch implementation of the Adam [51] optimizer with default hyperparameters. Batch sizes of 128 were used for all experiments.

Hyperparameters for the STG layer were set as described in Yamada et al. [23]. That is, we set the $\sigma$ for the $\mathcal{N}(0, \sigma^2)$ distribution used for drawing noise to $\sigma = 0.5$, and the regularization parameter $\lambda$ was tuned so that the desired number of gates were left open (i.e., $\mu_i > 0$) after convergence.

We reimplemented the unsupervised concrete autoencoder (CAE), differentiable unsupervised feature selection (DUFS), and stochastic gates (STG) baselines in PyTorch. Similar to CFS, for the CAE's reconstruction function we used a multilayer perceptron with two hidden layers of size 512. As in Balın et al. [3], we optimized the CAE using Adam [51] with default hyperparameters and set the temperature of the CAE's concrete selector layer following Balın et al. [3]: i.e., at the start of training, we set the temperature of the concrete layer to 10, and gradually annealed the temperature to a final value of 0.1 using the exponential decay schedule described in Appendix A. As in [3], we trained our concrete autoencoders until the mean of the concrete samples exceeded 0.99.

As in Lindenbaum et al. [36], we trained DUFS with an SGD optimizer with learning rate of 1 using the authors' proposed "parameter-free" loss function (i.e., eq. 9 of Lindenbaum et al. [36]) until convergence. For a given number of desired features $k$, the $k$ features with the highest gate values were then selected as described in Lindenbaum et al. [36].

For the supervised STG baseline [23], we used two hidden layers of 512 units. To adapt STG, which was designed for fully supervised scenarios, to the contrastive analysis setting, we trained it on the binary classification problem of distinguishing target versus background points. We optimized STG using Adam with default hyperparameters. For all experiments with STG, the regularization parameter $\lambda$ was tuned such that STG selected the desired number of features.

For all experiments with the supervised LassoNet model [34], we used the authors' Python package[5]. As with STG, we adapted the supervised LassoNet model to the contrastive analysis setting by training it on the binary classification problem of distinguishing target versus background points. The regularization parameter $\lambda$ was tuned to select the desired number of features, and all other hyperparameters were left at their default values in the LassoNet package.

---

[4]https://github.com/PyTorchLightning/pytorch-lightning
[5]https://github.com/lasso-net/lassonet

# E  Dataset descriptions

Here we provide additional details on the real-world datasets considered in our empirical evaluation of CFS in the main text. We also summarize these datasets in Table S1. Our code for preprocessing these datasets can be found at `www.placeholder.com`.[6]

**Mice protein expression**   This dataset [43, 52] consists of protein expression levels from healthy control mice and mice that have developed Down syndrome. Each mouse was injected with memantine, subjected to shock therapy, received both of these treatments, or none of them. Our goal here was to classify mice with Down syndrome based on their treatment regimens. We used data from healthy mice that did not receive any treatment as a background. As done in previous work (e.g., [3]), features were min-max normalized. Raw data was downloaded from `https://archive.ics.uci.edu/ml/machine-learning-databases/00342/`.

**Epithelial cell infection response**   We constructed our target dataset by combining two sets of gene expression measurements from Haber et al. [44]. These datasets consist of gene expression measurements of intestinal epithelial cells from mice infected with either *Salmonella* or *Heligmosomoides polygyrus (H. poly)*. Here our goal is to separate cells by infection type. As a background dataset we used measurements collected from healthy cells released by the same authors. As is standard for single-cell gene expression measurements (see e.g., [53] for a review), raw count values were library-size-normalized to account for differences in sequencing depth across cells. Normalized counts $x$ were then transformed according to $\log(x + 1)$. All preprocessing steps for this dataset were carried out using scanpy [54], a standard Python package for analyzing single-cell RNA-seq data. Gene expression count files were downloaded from the NIH Gene Expression Omnibus at `https://www.ncbi.nlm.nih.gov/geo/query/acc.cgi?acc=GSE92332`.

**AML treatment**   Here we combined datasets from Zheng et al. [45] containing single-cell gene expression measurements from two patients with acute myeloid leukemia (AML) before and receiving a blood cell transplant. Our goal here is to learn a representation that separates pre and post transplant measurements. As a background dataset we used expression measurements from two healthy control patients that were collected as part of the same study. Raw counts were normalized as described above for the epithelial cell infection response dataset. Files containing measurements from the first patient pre- and post-transplant can be found at `https://cf.10xgenomics.com/samples/cell-exp/1.1.0/aml027_pre_transplant/aml027_pre_transplant_filtered_gene_bc_matrices.tar.gz` and `https://cf.10xgenomics.com/samples/cell-exp/1.1.0/aml027_post_transplant/aml027_post_transplant_filtered_gene_bc_matrices.tar.gz`, respectively; from the second patient pre- and post-transplant at `https://cf.10xgenomics.com/samples/cell-exp/1.1.0/aml035_pre_transplant/aml035_pre_transplant_filtered_gene_bc_matrices.tar.gz` and `https://cf.10xgenomics.com/samples/cell-exp/1.1.0/aml035_post_transplant/aml035_post_transplant_filtered_gene_bc_matrices.tar.gz`, respectively; and from the two healthy control patients at `https://cf.10xgenomics.com/samples/cell-exp/1.1.0/frozen_bmmc_healthy_donor1/frozen_bmmc_healthy_donor1_filtered_gene_bc_matrices.tar.gz` and `https://cf.10xgenomics.com/samples/cell-exp/1.1.0/frozen_bmmc_healthy_donor2/frozen_bmmc_healthy_donor2_filtered_gene_bc_matrices.tar.gz`.

**ECCITE-seq perturbation screen**   ECCITE-seq [55] is a technique that allows for multimodal readouts of single-cell perturbation screens. Here we obtained an ECCITE-seq dataset from Papalexi et al. [14] that measured how various gene knockout perturbations affected the gene expression profiles of cells from the THP1 cancer cell line. Our goal here was to classify cells into one of three clusters of perturbations with similar effects identified by Papalexi et al. [14]. As a background dataset we used measurements from control cells with no perturbation collected as part of the same study. Raw gene expression counts were normalized as described above for the epithelial cell infection response dataset. Gene expression count files and associated metadata were downloaded from the NIH Gene Expression Omnibus at `https://www.ncbi.nlm.nih.gov/geo/query/acc.cgi?acc=GSE153056`.

---

[6]Code available in supplementary materials and will be made public upon acceptance.

Table S1: Summary of real-world datasets considered in the main text.

| Dataset | Features ($d$) | Train size (target) | Train size (background) | Test size | Classes |
|---|---|---|---|---|---|
| Grassy MNIST | 784 | 48,000 | 48,000 | 12,000 | 10 |
| Mice Protein | 77 | 444 | 135 | 111 | 4 |
| Epithelial cell | 15,215 | 3,584 | 3,240 | 897 | 2 |
| AML treatment | 12,079 | 9,919 | 4,457 | 2,480 | 2 |
| ECCITE-seq | 18,649 | 14,674 | 2,386 | 3,669 | 3 |

# F    Deeper analysis of Epithelial Cell dataset results

To illustrate how the features selected by CFS may lead to new biological insights, we performed an additional analysis of the specific gene features selected by CFS versus supervised methods for the mice intestinal epithelial cell infection dataset from Haber et al. [44]. In this dataset gene expression levels were compared between control cells (background) and cells exposed to either the bacteria *Salmonella*or the parasite *H. poly* (target). For this analysis we compared the features selected by CFS and STG, the best-performing supervised baseline in our quantitative experiments, with the number of features $k$ set to 20.

We first considered the overlapping genes selected by both methods, and we found that both methods selected genes involved in the inflammatory response (e.g. *Cd74*) and previously studied defense responses to pathogens (e.g. *Reg3b*, *Reg3g*). We next considered the genes selected only by CFS and not by STG. We found that CFS tended to select genes (Figure S5a) with clear differences in expression patterns between the two pathogens (e.g. *Gm42418*), including a number of enterocyte marker genes (e.g. *Apoa1*, *Guca2b*, *Fabp1*) that were upregulated in *Salmonella* cells compared to *H. poly* cells. Notably, the enterocyte markers exhibited a clear bimodal pattern in their expression levels for *Salmonella*-infected cells.

Upon further inspection we found that the observed upregulation of these genes was due to a combination of two phenomena. First, as noted in Haber et al. [44], *Salmonella*-infection resulted in an increase in the number of enterocytes relative to healthy and *H. poly*-infected populations. Despite the increase in the number of enterocytes, the distribution of expression levels for these marker genes were notably similar for control and *Salmonella* enterocytes (Figure S5b). On the other hand, we found (Figure S5b) that these genes' expression levels were substantially upregulated in *Salmonella*-infected non-enterocytes compared to control non-enterocytes. This cell-type-specific response was not noted in Haber et al. [44], and this finding thus illustrates the potential for CFS to uncover insights that may be missed with standard workflows.

On the other hand, the genes selected by STG (but not CFS) consisted of immune response genes (e.g. *Lgals9*, *Krt19*) with altered expression compared to controls, but which did not exhibit other notable patterns among the perturbed cells (Figure S5c). These results indicate that supervised methods may select features that can distinguish target vs. background data, but which do not capture more subtle phenomena within the target data.

# G  Supplementary Figures

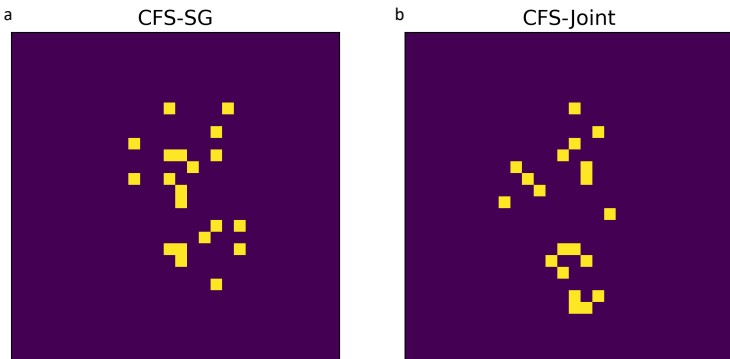

Figure S1: Qualitative results for stop-gradient (SG) and joint training CFS variants on the Grassy MNIST dataset for $k = 20$ features.

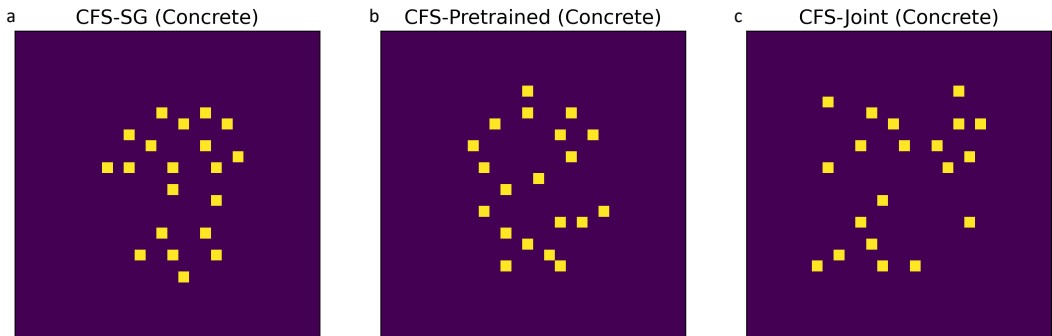

Figure S2: Qualitative results for CFS implemented with the concrete selection layer of Balın et al. [3] on the Grassy MNIST dataset for $k = 20$. We find that the gradient stopping (GS) and pretrained CFS variants select features concentrated towards the center as desired, while features selected by the joint training CFS variant are more spread out away from the center.

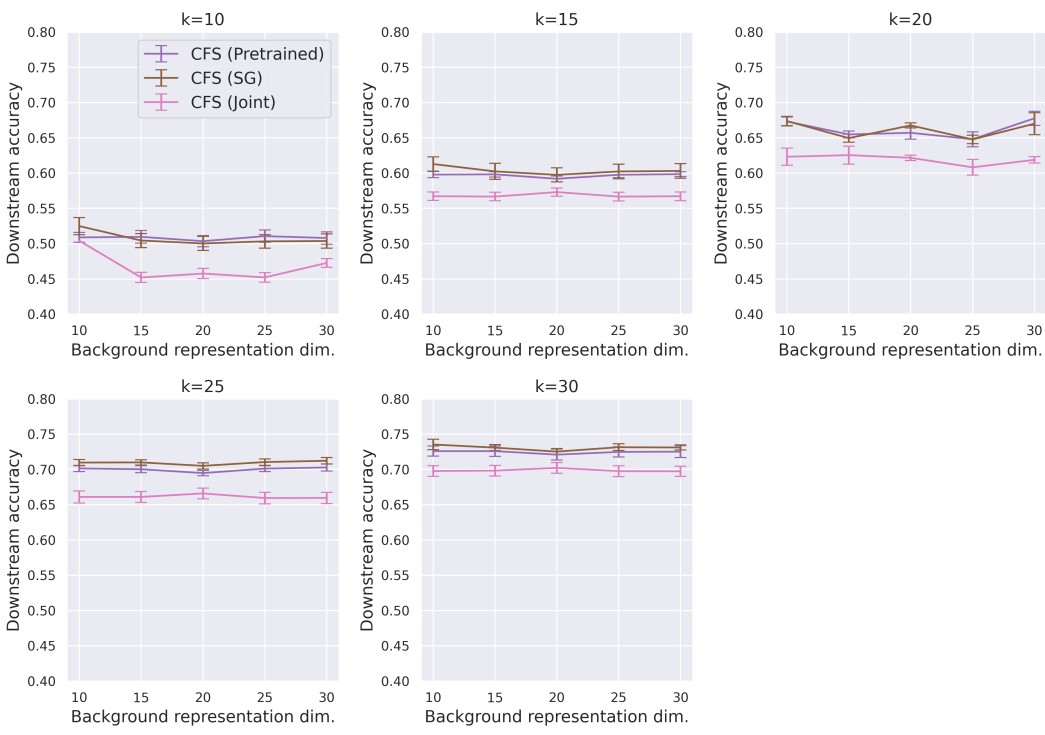

Figure S3: Varying background representation dimension for various numbers of selected features $k$ on Grassy MNIST.

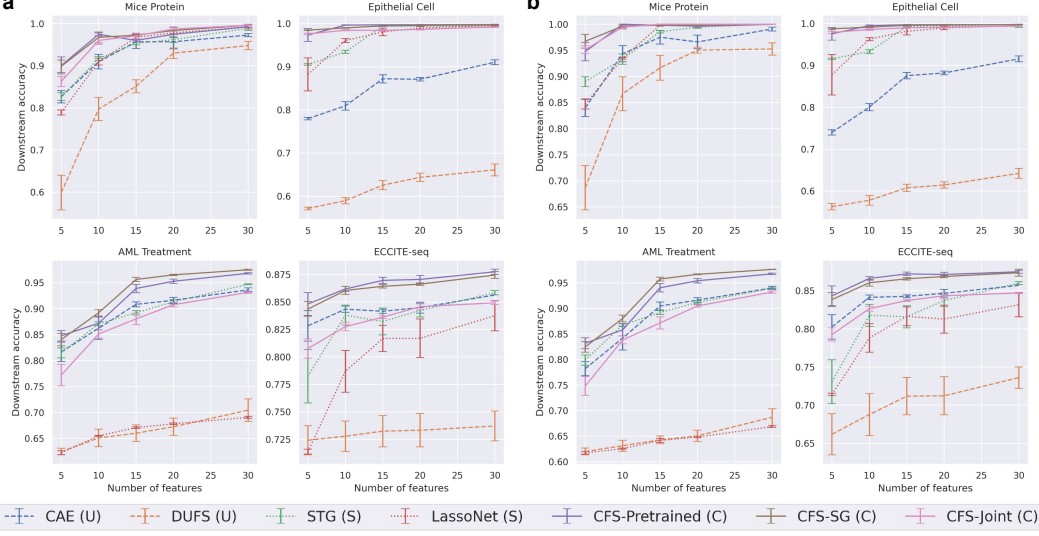

Figure S4: Downstream classification results using (**a**) XGBoost and (**b**) neural networks as the downstream classifier.

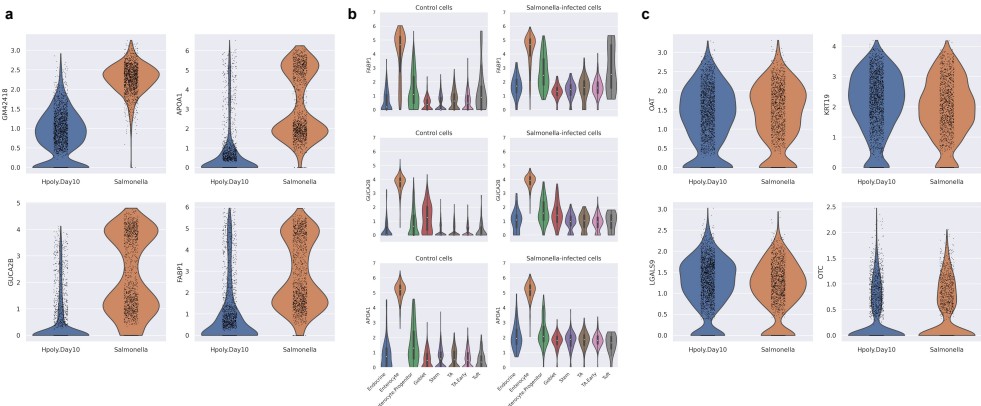

Figure S5: **(a)** Sample of gene features selected by CFS. **(b)** Visualizing cell-type-specific changes in gene expression in response to *Salmonella* infection for genes selected by CFS. **(c)** Sample of genes selected by the best performing supervised feature selection method STG.

# H Supplementary Tables

Table S2: Assessing the quality of features selected by CFS for varying numbers of background samples on the mice protein dataset for $k = 5$ features. Feature quality was assessed by training a random forest classifier to distinguish between known groups of target samples, and we report the mean accuracy $\pm$ standard error over five random seeds on a held out test set. We find that feature quality initially increases as the number of samples increases, with saturation achieved around 50 background samples for this dataset.

| Number of Background Samples | CFS (Pretrained) | CFS (GS) |
|:---:|:---:|:---:|
| 10 | $0.902 \pm 0.021$ | $0.902 \pm 0.021$ |
| 20 | $0.912 \pm 0.012$ | $0.912 \pm 0.012$ |
| 30 | $0.920 \pm 0.014$ | $0.922 \pm 0.012$ |
| 50 | $0.934 \pm 0.012$ | $0.933 \pm 0.014$ |
| 70 | $0.932 \pm 0.009$ | $0.936 \pm 0.012$ |
| 135 | $0.935 \pm 0.011$ | $0.935 \pm 0.015$ |

Table S3: Comparing the quality of features selected by the CFS architecture with a fixed, randomly initialized background encoder (CFS (Random)) versus our pretrained and gradient stopping (GS) CFS variants on the mice protein dataset. For comparison we also include the results from STG, our best performing non-contrastive baseline. Feature quality was assessed by training a random forest classifier to distinguish between known groups of target samples, and we report the mean accuracy $\pm$ standard error over five random seeds on a held out test set. We find that CFS (Random) significantly underperforms our pretrained and GS CFS variatns for lower numbers of features, with comparable performance to STG.

| Number of Features | CFS (Random) | STG | CFS (Pretrained) | CFS (GS) |
|:---:|:---:|:---:|:---:|:---:|
| 5 | $0.888 \pm 0.004$ | $0.865 \pm 0.006$ | $0.935 \pm 0.011$ | $0.935 \pm 0.015$ |
| 10 | $0.918 \pm 0.003$ | $0.923 \pm 0.007$ | $0.993 \pm 0.005$ | $0.989 \pm 0.007$ |
| 15 | $0.970 \pm 0.002$ | $0.968 \pm 0.004$ | $0.995 \pm 0.004$ | $0.993 \pm 0.007$ |
| 20 | $0.985 \pm 0.005$ | $0.980 \pm 0.005$ | $0.989 \pm 0.009$ | $0.998 \pm 0.002$ |
| 30 | $0.995 \pm 0.001$ | $0.991 \pm 0.003$ | $0.998 \pm 0.002$ | $0.996 \pm 0.002$ |

