# OpenReview forum: "Feature Selection in the Contrastive Analysis Setting"
_NeurIPS.cc/2023/Conference — NeurIPS 2023 poster_

### Official Review · Reviewer_nNUw · 2023-06-19

**Soundness:** 3 good
**Presentation:** 3 good
**Contribution:** 2 fair
**Rating:** 4
**Confidence:** 2

**Summary:**

The paper studied feature selection in the contrastive analysis setting. There are two types of datasets, the background dataset and the target dataset. The background dataset contains variations due to the uninteresting sources. The target dataset contains observations with variability from both ``interesting'' and ``uninteresting'' sources. The goal of the problem is to select interesting variables. Since interesting variables only exist in target variable, the feature selection is performed on the target dataset. To make use of the background dataset, the embedding parameters learned by the auto encoder on the background dataset is shared with the target dataset in feature selection.

The problem is with practice interest. But the exact layers of background representation encoder, the background reconstruction and the target reconstruction network are not specified. The paper vaguely mentioned that it is similar to auto-encoder. Since the architecture is the main contributions of the paper, it is better to specify the networks.

The theoretical analysis is about mutual information $I$, which is related with multiple variables. However, $I$ is not defined in the paper.

Feature selection selects features that are most related with the target. However, the experiments are based on MNIST. There is no analysis about what features are selected. Though the motivation of the problem is about biomedical study, there is no experiments on related area.

**Strengths:**

The paper is well-written. The motivation is strong. The paper proposed a method, made related analysis and conducted some experiments.


**Weaknesses:**

1. The architecture is not clear to me, specially the representation / reconstruction networks.

2. There is no ablation study if there is no background dataset, the background representation encoder is initialized by random distribution.

3. There is no analysis about the impact of different population of background dataset and target dataset.

4. There is not enough experiments to validate the performance of the proposed method.

**Questions:**

1. What is the definition of the mutual information $I$?
2. What is the impact of the population of background dataset and target dataset?
3. What does the networks $g$, $h$ and $f$ look like?
4. Is the feature space of the background dataset is a subset of the feature space of the target dataset?
5. What is the scale of population and feature dimension to make the proposed method work?

**Limitations:**

1. The theoretical analysis is not clear which means the performance is not guaranteed.
2. Assume the background dataset is in large scale, how to select the background dataset related to the target dataset is important but not discussed here.
3.There are many assumptions to implement the algorithms in real-world applications, such as the population between background and target datasets, the number of selected features $s$ compared with the global feature space.

---

> ### Author Rebuttal · Authors · 2023-08-09
>
> We thank the reviewer for examining our work and providing their feedback. We’re thrilled to hear that the reviewer found our work “well-written” with “strong” motivation. In the space below we respond to the reviewer’s specific concerns:
>
> **Evaluations beyond MNIST:** The reviewer noted that, despite our method being motivated by biomedical applications, our evaluation of CFS consisted solely of experiments on MNIST. **We believe that the reviewer’s claim here results from a misunderstanding.** In Figure 5 of our submission, we provide results from applying CFS and baseline methods to **four real-world biomedical datasets**, including three single-cell RNA sequencing datasets and a proteomics dataset. In these experiments, we found that CFS consistently outperformed baseline methods.
>
> **Definition of mutual information (MI):** We thank the reviewer for pointing out this omission. The MI at an intuitive level quantifies the amount of information obtained about the value of one random variable $X$ from observing the value of a second variable $Y$. More formally, for two discrete random variables $X$ and $Y$, the MI $I(X; Y)$ is defined as:
>
> $$ I(X; Y) = \sum_{y \in \mathcal{Y}} \sum_{x \in \mathcal{X}} P_{X,Y}(x, y)\log\frac{P_{X,Y}(x,y)}{P_{X}(x)P_{Y}(y)} $$
>
> where $P_{X,Y}$ is the joint probability mass function of $X$ and $Y$ and $P_{X}$ and $P_{Y}$ are the marginal probability mass functions of $X$ and $Y$.
>
> In our revised manuscript we will include an intuitive description of the mutual information, as well as a formal definition with references to texts (e.g. Cover and Thomas [1]) that provide a deeper introduction to the subject.
>
> **Feature space of target vs. background dataset:** We thank the reviewer for raising this potential point of confusion. CFS (and CA methods more generally) require that the feature space for target and background samples are identical. For example, in the single-cell RNA sequencing datasets considered in our study the same genes were measured for background and target samples. We will clarify this in the final manuscript.
>
> **Required scale of population and feature dimension:**  We thank the reviewer for raising this point. We found empirically that CFS worked across a variety of population and feature dimension sizes. For example, we found that CFS achieved state of the art results on both low-dimensional datasets (e.g. the mice protein dataset with 77 features) as well as the higher dimensional datasets (e.g. the ECCITE-seq dataset with 18,649 features). Similarly, we found that CFS achieved state of the art results across datasets with varying numbers of samples, from hundreds (mice protein) to thousands (epithelial cell and AML treatment) to tens of thousands of samples (Grassy MNIST, ECCITE-seq). These results thus suggest that CFS is effective across a variety of feature dimensions and dataset sizes.
>
> **Ablation without background dataset:** As suggested by the reviewer, we computed the downstream classification of features selected by CFS with a randomly initialized encoder. We provide the results for the mice protein dataset below, and we will include additional results in the supplementary materials for the final manuscript. We found that randomly initializing the encoder led to a significant drop in performance for lower numbers of features. Notably, we found that CFS with random initialization had comparable performance to the supervised STG method (results included below); this is not entirely surprising, as CFS with a randomly initialized background encoder is nearly identical to the STG method.
>
> | k | CFS (Random) | STG | CFS (Pretrained)  | CFS (GS) |
> |---|---|---|---|---|
> | 5 | $0.888 \pm 0.004$ | $0.865 \pm 0.006$ | $0.935 \pm 0.011$ | $0.935 \pm 0.015$ |
> | 10 | $0.918 \pm 0.003$ | $0.923 \pm 0.007$ | $0.993 \pm 0.005$ | $0.989 \pm 0.007$ |
> | 15 | $0.97 \pm 0.002$  | $0.968 \pm 0.004$ | $0.995 \pm 0.004$ | $0.993 \pm 0.007$ |
> | 20 | $0.985 \pm 0.005$ | $0.980 \pm 0.005$ | $0.989 \pm 0.009$ | $0.998 \pm 0.002$ |
> | 30 | $0.995 \pm 0.001$ | $0.991 \pm 0.003$ | $0.998 \pm 0.002$ | $0.996 \pm 0.002$ |
>
> Similar results were obtained for other datasets and will be included in the final manuscript.
>
> **Impact of target versus background dataset size:** We first note that the datasets we considered in our manuscript had a variety of background to target dataset size ratios, and CFS consistently outperformed our baselines on these datasets. For example, the mice protein dataset had 135 background to 444 target samples, while the epithelial cell dataset had a nearly 1 to 1 target to background ratio; we refer the reviewer to Appendix E of the supplementary materials for full details for each dataset. In addition, to better understand CFS's performance with respect to different numbers of target vs. background samples, we assessed the quality of features selected by CFS on the mice protein dataset with $k = 5$ for varying numbers of background samples, where 135 is the full number of background samples (and this dataset contained 435 target samples):
>
> | Num background samples | CFS (Pretrained) | CFS (GS) |
> |:---:|:---:|:---:|
> | 10 | $0.902 \pm 0.021$ | $0.902 \pm 0.021$ |
> | 20 | $0.912 \pm 0.012$ | $0.912 \pm 0.012$ |
> | 30 | $0.920 \pm 0.014$ | $0.922 \pm 0.012$ |
> | 50 | $0.934 \pm 0.012$ | $0.933 \pm 0.014$ |
> | 70 | $0.932 \pm 0.009$ | $0.936 \pm 0.012$ |
> | 135 | $0.935 \pm 0.011$ | $0.935 \pm 0.015$ |
>
> We observed that CFS's performance improved initially as more background samples were introduced, with saturation achieved at ~50 background samples. These results provide further evidence that CFS can select high quality features even when the number of background samples is small compared to the number of target samples (435 in this case). Similar results were obtained for other datasets with varying values of $k$ and will be included in the final manuscript.
>
> [1]: Cover, Thomas M. Elements of information theory. John Wiley & Sons, 1999.

---

> > ### Comment · Reviewer_nNUw · 2023-08-21
> >
> > Thanks to the authors for addressing some of my concerns. Thank you for including the experimental results. When k reaches 20, the difference between the performance of CFS (random) and CFS (pretrained) is trivial. Does it mean pretraining is not necessary when k reaches >20? Is it expected or is it related with the pretraining network?
> >
> > With the number of background samples increase, we expected the performance of CFS (pretrained) improved as well. The performance of CFS (pretrained) and CFS (GS) is close. Is that aligned with the design of the proposed method?

---

> > > ### Author Response · Authors · 2023-08-21
> > >
> > > We thank the reviewer for their feedback, and we're thrilled to hear that our rebuttal addressed some of your concerns. Responses to the reviewer's remaining questions are below. Please let us know if we can clarify any remaining points before the end of the discussion period.
> > >
> > > **Re: CFS (random) vs CFS (pretrained) performance for $k \geq 20$:** We thank the reviewer for raising this potential point of confusion and allowing us to clarify. We believe that the observed phenomenon (i.e., CFS (Random) having comparable performance to CFS (Pretrained) and CFS (GS) for $k \geq 20$) is due to the low dimensionality of the mice protein dataset (only 77 total features). Thus, with $k \geq 20$ a significant proportion of the total feature space is being selected for this dataset, and it is relatively easy for CFS (Random) and other baseline methods to achieve good performance on the corresponding downstream classification task. Indeed, as shown in Figure 5 in the main text, fully supervised baseline methods are also able to achieve near perfect performance for $k \geq 20$ on this dataset.
> > >
> > > On the other hand, for higher-dimensional datasets it is more likely that CFS (Random), which does not fully leverage the background dataset, will select lower-quality features than CFS (Pretrained) or CFS (GS) even with $k \geq 20$. To illustrate this point, below we provide results on the remaining real-world datasets for CFS (Random) and our proposed CFS (Pretrained) and CFS (GS) methods for $k=20$. We note that these dataset have significantly higher dimensionality, with all having over 10,000 features. **We find that CFS (Random) indeed consistently underperforms our proposed CFS (Pretrained) and CFS (GS) methods on these higher-dimensionality datasets with $k=20$.**
> > >
> > > | Dataset | CFS (Random) | CFS (Pretrained) | CFS (GS) |
> > > | --- | --- | --- | --- |
> > > | ECCITE-seq | $0.796 \pm 0.011$ | $0.871 \pm 0.003$ | $0.868 \pm 0.001$ |
> > > | AML | $0.912 \pm 0.015$ | $0.950 \pm 0.004$ | $0.962 \pm 0.001$ |
> > > | Epithelial cell | $0.954 \pm 0.029$ | $0.997 \pm 0.001$ | $0.996 \pm 0.001$ |
> > >
> > >
> > > **Re: similar performance of CFS (Pretrained) and CFS (GS):** To clarify, we designed both CFS (Pretrained) and CFS (GS) to optimize for the same objective. In particular, both train a background representation network designed to exclusively capture the (uninteresting) variations shared across target and background datasets, and the background representations are then leveraged to select specifically the $k$ features that best reflect target-data-specific variations. Within this framework, the pretraining and gradient stopping (GS) strategies are simply two potential mechanisms for ensuring that the background representation network indeed exclusively captures uninteresting variations. Thus, it is indeed not surprising that CFS (Pretrained) and CFS (GS) exhibited similar performance.

---

> > > > ### Comment · Reviewer_nNUw · 2023-08-21
> > > >
> > > > Thank authors for clarification and more experimental results.
> > > > CFS (random) and CFS (pretrained) performed equally well when k reaches 20, in the case that the global feature space is 77. And the extra experimental results are conducted on datasets ECCITE-seq, AML,and  Epithelial cell which have higher original feature space (~10,000 features), the intuitive idea is that when k reaches certain threshold (depending on the original feature space), the difference between the performance of CFS (random) and CFS pretrained could be trivial.
> > > >
> > > > The  original question is what is the impact of the target versus background dataset size towards the performance of the proposed methods. The conclusion is that there is no major impact on the proposed methods CFS (pretrained) and CFS (GS). Hence, I do not see the necessary of proposing two methods that perform similarly.
> > > >
> > > > Thanks for the all the efforts by the authors. I will keep my score as it is.

---

> > > > > ### Author Response · Authors · 2023-08-21
> > > > >
> > > > > Dear reviewer,
> > > > >
> > > > > Thank you again for your feedback. Before the end of the discussion period, we just wanted to clarify a couple of final points.
> > > > >
> > > > > **Re: Difference between CFS (Pretrained) and CFS (GS) versus CFS (Random) for higher values of $k$:** The reviewer is correct that for sufficiently large numbers of selected features $k$ that the performance of CFS (Random) and the CFS (Pretrained) and CFS (GS) methods will likely become similar (for example, this is trivially true when $k$ is equal to the dimensionality of the full feature space). However, we emphasize that it is often the case that the feature budget $k$ is restricted to some small value and cannot be arbitrarily large. This may be due to interpretability concerns, computational or storage constraints, or may be due to technological limitations that only permit the collection of a small number of features. For example, fluorescence in situ hybridization (FISH) technologies allow biologists to measure gene expression levels at extremely high precision in single cells; however, due to technological limitations only a small subset of genes may be measured, and thus methods for pre-selecting informative subsets of genes are required before running a FISH experiment [1]. **In these realistic scenarios with low values of $k$ relative to the size of the full feature space, we found that CFS (Pretrained) and CFS (GS) consistently outperformed CFS (Random).**
> > > > >
> > > > > **Re: CFS (Pretrained) versus CFS (GS):** To clarify, in our manuscript **we are proposing one key idea** (i.e., isolating shared variations in the background representation space so that the $k$ selected features best reflect target-specific-variations), **and both CFS (Pretrained) and CFS (GS) are implementations of this same single idea**. In particular, CFS (Pretrained) explicitly performs the two-step optimization procedure described in Section 4 and depicted in Figure 2. On the other hand, CFS (GS) implicitly accomplishes the same goal by making use of the gradient stopping operation, thus potentially simplifying implementation by avoiding the need for an explicit pretraining step. As both CFS (Pretrained) and CFS (GS) achieve similar performance, a potential user may thus select either variant depending on which they find easier to implement, which we see as a strength of our work.
> > > > >
> > > > > [1]: Covert, I., Gala, R., Wang, T. et al. Predictive and robust gene selection for spatial transcriptomics. Nat Commun 14, 2091 (2023). https://doi.org/10.1038/s41467-023-37392-1

---

### Official Review · Reviewer_SoSQ · 2023-07-03

**Soundness:** 2 fair
**Presentation:** 1 poor
**Contribution:** 2 fair
**Rating:** 7
**Confidence:** 3

**Summary:**

 The paper aims to solve the problem of “salient” feature selection in a Contrastive Analysis setting, termed “Contrastive Feature Selection.” The authors' primary focus lies in uncovering distinctive features that highlight salient variations present in a specific target dataset that are absent in a standard background dataset. The paper discusses & compares the limitations of popular feature selection methods both in supervised & unsupervised methods settings and further attempts to formulate a “Contrastive Feature Selection” method to extract those salient features. The paper provides an initial empirical formulation based on intuitions, then formalizes the idea using principles of Information Theory (bounds on Mutual Information, Entropy)  and finishes with empirical results on semi-synthetic (vision) and real-world (medical) datasets. Overall, the paper has made a novel attempt at feature selection in a weak supervision setting using a background and target dataset, with the goal being to learn features specific to the target dataset.

**Strengths:**

1. The paper introduces a "contrastive feature selection" method. The authors provide experimental validation with popularly used baselines (both supervised+unsupervised)  employed in feature selection.
2. The paper also provided intuitive, experimental, and mathematical proofs (based on Information Theoretic bounds) to justify their proposed method.
3.  Another strength of the proposed approach is it relies only on weak supervision (whether a data sample is part of a background/target dataset).
4. The results show the superior performance of the proposed feature selection method for downstream tasks.
5. Code provided for reproducibility.

**Weaknesses:**

1. The authors use only "Random Forest Classifiers" for their downstream tasks. It would be a good idea to add comparative results (perhaps in the Supplementary) using other popular classifiers (like Decision Trees, SVM, ANNs, XGBoosts, etc.) to show that the feature selection works across different classifiers.
2. The paper shows empirical results on the "Grassy-MNIST" and four other medical datasets. However, results on more practical classification tasks (>10 classes) would make more sense. Also, it would be interesting to see the performance variation for a particular dataset like GrassyMNIST as background vs. target dataset size is varied.
3. A more extensive ablation of varying dimensions of the latent {k,l} is missing.
4. The overall organization of the paper can be improved. It feels a bit unorganized going through the paper.


**Questions:**

1. Is adding experiments based on Weakness Section Points 1-2 possible?
2. Also, is it possible to provide a theoretical explanation or intuitive insight why the two variants of CFS-SG and CFS-Pretrained result in a similar performance, but the top performer changes when tested across datasets?  Is it related to some factor like dataset sizes/ number of classes for classification?


**Limitations:**

1. Limited evaluation of real-life classification datasets. All datasets had <= ten classes.
2. Validation of the selected feature performance when using different classifier models.

---

> ### Author Rebuttal · Authors · 2023-08-08
>
> We thank the reviewer for examining our work and providing their feedback. We’re thrilled to hear that the reviewer found that CFS represents a “novel attempt at feature selection” and that our results  show “the superior performance of the proposed feature selection method”. We have responded to general concerns shared across reviewers in our main response, and In the space below we respond to the reviewer’s specific concerns.
>
> **Additional downstream classifiers:** As requested by the reviewer, we have re-computed our downstream classification metrics using two alternative classifiers: XGBoost and neural networks, and we provide the results for the real-world biomedical datasets in **Fig. 1** in the one-page rebuttal PDF. We found that the results with these new classifiers were largely consistent with the original results using random forests, and we will include these new results in the final version of our supplementary materials.
>
> **Additional ablations of background representation dimension vs. number of features:** We thank the reviewer for this suggestion. To satisfy the reviewer’s concerns, we performed further experiments on the Grassy MNIST dataset measuring CFS’s performance (as measured by downstream classification accuracy for the selected features) for each combination of background representation dimension $l \in [10, 15, 20, 25, 30]$ and number of selected features $k \in [10, 15, 20, 25, 30]$.  We present these results in **Fig. 2** in the rebuttal PDF document. We found that, for a given number of target features $k$, performance was largely consistent with respect to the choice of background representation dimension $l$.
>
> **Evaluations on "more realistic" (>10 classes) datasets:** We thank the reviewer for raising this point. Despite our efforts, we were unable to find a dataset that is both suitable for contrastive analysis (i.e., has an obvious target versus background) and which contains more than 10 ground truth class labels. However, we emphasize that even for datasets where the downstream classification task should be relatively “easy” (e.g. only two ground truth classes), CFS often substantially outperformed supervised and unsupervised baselines. For example, the features selected by the pretrained and gradient stopping CFS variants for the epithelial cell dataset with $k = 5$ led to downstream classification accuracy > 97.5%, while the next best baseline model achieved 91.8%.
>
> Moreover, to further demonstrate the utility of CFS in real-world scenarios, in the final manuscript we will include new results from a detailed case study analyzing the features selected by CFS when applied to the mice intestinal epithelial cell infection dataset from Haber et al. [1] (see response to reviewer **xS67** for details). We found that the features selected by CFS reflected cell-type-specific responses to infections that were not discussed in the original study. This result thus suggests that CFS may aid in the exploration of realistic/real-world datasets compared to standard analysis workflows.
>
> **When to choose pretraining versus gradient stopping?:** We thank the reviewer for raising this point. We do not have a precise explanation for when pretraining versus gradient stopping should perform better for different datasets; however, we note that performance was usually comparable between both CFS variants (i.e., overlapping error bars). We also note that from a user’s perspective, this choice may simply be regarded as a hyperparameter to be tuned e.g. based on performance on a validation set. For example, the user could compare whether the features selected by pretraining or gradient stopping lead to better reconstruction error on validation data.
>
> **Improved organization of the paper:** We thank the reviewer for their constructive criticism re: the organization.  To improve the flow of the manuscript, we will add a more detailed outline of the organization of our paper to the introduction, as well as improve transitions between sections. We are happy to include any other specific suggestions on this front for the final version of our manuscript.
>
> **Impact of target versus background dataset size:** We first note that the datasets considered in our manuscript had a variety of background to target dataset size ratios, and CFS consistently outperformed baseline methods on all of these datasets. For example, the mice protein dataset had 135 background samples to 444 target samples, while the epithelial cell dataset had a nearly 1 to 1 target to background sample ratio; we refer the reviewer to Appendix E of the supplementary materials for full details for each dataset. In addition, to better understand CFS's performance with respect to different numbers of target versus background samples, we assessed the quality of features selected by CFS on the mice protein dataset with $k = 5$ for varying numbers of background samples, where 135 is the full number of background samples (and this dataset contained 435 target samples):
>
> | Num background samples | CFS (Pretrained) | CFS (GS) |
> |:---:|:---:|:---:|
> | 10 | $0.902 \pm 0.021$ | $0.902 \pm 0.021$ |
> | 20 | $0.912 \pm 0.012$ | $0.912 \pm 0.012$ |
> | 30 | $0.920 \pm 0.014$ | $0.922 \pm 0.012$ |
> | 50 | $0.934 \pm 0.012$ | $0.933 \pm 0.014$ |
> | 70 | $0.932 \pm 0.009$ | $0.936 \pm 0.012$ |
> | 135 | $0.935 \pm 0.011$ | $0.935 \pm 0.015$ |
>
> We observed that CFS's performance improved initially as more background samples were introduced, with saturation achieved at ~50 background samples. These results provide further evidence that CFS can select high quality features even when the number of background samples is small compared to the number of target samples (435 in this case). Similar results were obtained for other datasets with varying values of $k$, and these will be included in the updated supplementary materials.
>
> [1]: Haber, A.L., et al. "A single-cell survey of the small intestinal epithelium." Nature 551.7680 (2017): 333-339

---

> > ### Comment · Reviewer_SoSQ · 2023-08-15
> >
> > Thanks to the authors for addressing my concerns.
> >
> > Based on the reviews from the other reviewers and the related rebuttal clarifications along with the rebuttal document provided, I am happy to upgrade my score.

---

> > > ### Author Response · Authors · 2023-08-15
> > > **Thank you for engaging in the discussion period!**
> > >
> > > Dear Reviewer SoSQ,
> > >
> > > Thank you for engaging in the discussion period. We're thrilled to see that you've decided to upgrade your score! Please let us know if you have any further questions before the end of the discussion period.

---

### Official Review · Reviewer_xS67 · 2023-07-05

**Soundness:** 3 good
**Presentation:** 3 good
**Contribution:** 3 good
**Rating:** 7
**Confidence:** 4

**Summary:**

The authors tackle the important problem of feature selection in a contrastive setting. Supervised and unsupervised feature selection problems have been extensively studied and have many high-impact applications. In this work, the authors design an algorithm for finding features that are more enriched in a target set than a background set (contrast). This type of comparison is widely used in biology, for example, in differential expression gene (DEG) analysis, where biologists look for differentially expressed genes across different medical conditions. The proposed method is based on a two-step procedure; in the first step, an autoencoder is trained to learn the latent information required to represent the background data. In the second step, a feature selection module is trained to identify the set of features required to capture the additional information needed to represent the target data. To sparsify the feature selection module, they use the recently proposed stochastic gates, which are continuously relaxed random variables. The new approach is motivation using a mutual information perspective. Several real and semi-synthetic datasets are used to evaluate the approach.

**Strengths:**

The problem is important with many applications in bioinformatics. The proposed is new and seems highly appropriate for the suggested problem. The experimental results highlight that the approach can lead to advantages compared to existing schemes. The authors propose a few variations for training the method and compare their quality. The benefits of the two-stage approach are justified using mutual information under mild assumptions.

**Weaknesses:**

Conceptually the problem could be solved using supervised feature selection; the authors also use supervised FS models as baselines. I understand that in terms of some applications, the setting is different. However, this is not well explained in the paper. Specifically, the authors should add information and results on how the selected features differ for the contrastive analysis case compared with the supervised (or unsupervised) case.  Accuracy is important, but interpretability is vital in the examples motivated by the authors. This could be done using synthetic data or by benchmarking the method (and others) on a previously studied DEG problem. This will significantly improve the paper and help increase its impact and use in bioinformatics.

**Questions:**

Some parts of the paper are very similar to [37], for example, two first sentences in the abstract and figure 1. Propper credit should be given here.
Was the 80-20 percent split performed for FS or only for the classification part? This is not clear from the text.
What are the architectures used for learning the background? Target? And also for STG and CAE?
How are those tuned?
While the authors demonstrate the merits of the approach for improving class separation (without access to class labels), I’m missing an evaluation that is tied to the motivation. Specifically, an example is that the approach is beneficial for DEG. As mentioned, this could be done using synthetic or real data with known gene markers. This would substantially improve the paper.


**Limitations:**

The limitations of the method are not discussed in the paper.

---

> ### Author Rebuttal · Authors · 2023-08-08
>
> We thank the reviewer for examining our work and providing their feedback. In the space below we respond to the reviewer’s specific concerns.
>
> **Train/test split:** The training points from the 80/20 test split were used to select features using CFS and train the subsequent classifiers, while the test points were only used to evaluate the performance of the final downstream classification models. We will clarify this  in the revised manuscript.
>
> **Clarifying the applications of contrastive analysis:** We thank the reviewer for pointing out the potential confusion on this point. To clarify, in the contrastive analysis setting a data analyst is interested in isolating patterns enriched in target samples compared to a corresponding set of background samples generated from “uninteresting” sources of variation. Previous works have shown that isolating these variations from confounding sources of variations shared with the background can lead to new biological insights, including in large-scale genetic perturbation screens [1, 2], and on autism spectrum disorder from MRI scans [3]. While previous works have focused on representation learning (e.g., via variational autoencoders) for CA, to our knowledge CFS is the first method designed for feature selection in the CA setting, which may provide increased interpretability.
>
> As done in our manuscript, fully supervised selection methods can in theory be naively applied to this problem by selecting features that discriminate between target and background points. However, as seen in our qualitative Grassy MNIST results in Figure 3 of the original manuscript, features that discriminate between the target and background often do not reflect the full spectrum of target-data-specific salient variations. Thus, methods designed to better exploit the weak supervision in CA may select features that reflect a wider variety of target-dataset-specific phenomena (e.g. fine-grained target-dataset-specific cellular states), and lead to additional insights.
>
> To illustrate this point, we provide the results of a deeper analysis of the features selected by CFS  for the mice intestinal epithelial cell infection dataset from Haber et al. [4] below, and we will make sure to clarify these nuances in the final manuscript.
>
> **Deeper analysis of features selected by CFS vs. supervised methods for epithelial cell dataset:** To illustrate how the features selected by CFS may lead to new biological insights, we performed an additional analysis of the specific gene features selected by CFS versus supervised methods for the mice intestinal epithelial cell infection dataset from [4]. In this dataset gene expression levels were compared between control cells (background) and cells exposed to either the bacteria Salmonella or the parasite H. poly (target). For this analysis we compared the features selected by CFS and STG, the best-performing supervised baseline in our experiments, with the number of features $k$ set to 20. We refer the reviewer to the rebuttal PDF for additional supporting figures.
>
> We first considered the overlapping genes selected by both methods, and we found that both methods selected genes involved in the inflammatory response (e.g. *Cd74*) and previously studied defense responses to pathogens (e.g. *Reg3b*, *Reg3g*). We next considered the genes selected only by CFS and not by STG. We found that CFS tended to select genes (**Fig. 4a** in the rebuttal PDF) with clear differences in expression patterns between the two pathogens (e.g. *Gm42418*), including a number of enterocyte marker genes (e.g. *Apoa1*, *Guca2b*, *Fabp1*) that were upregulated in Salmonella cells compared to H.poly cells. Notably, the enterocyte markers exhibited a clear bimodal pattern in their expression levels for Salmonella-infected cells.
>
> Upon further inspection we found that the observed upregulation of these genes was due to a combination of two phenomena. First, as noted in [4], Salmonella-infection resulted in an increase in the number of enterocytes relative to healthy and H.poly-infected populations. Despite the increase in the *number* of enterocytes, the distribution of expression levels for these marker genes were notably similar for control and Salmonella enterocytes (**Fig. 4b** in rebuttal PDF).  On the other hand, we found (**Fig. 4b** in PDF) that these genes’ expression levels were substantially upregulated in Salmonella-infected non-enterocytes compared to control non-enterocytes. This cell-type-specific response was not noted in [4], and this finding thus illustrates the potential for CFS to uncover insights that may be missed with standard workflows.
>
> On the other hand, the genes selected by STG (but not CFS) consisted of immune response genes (e.g. *Lgals9*, *Krt19*) with altered expression compared to controls, but which did not exhibit other notable patterns among the perturbed cells (**Fig. 4c** in rebuttal PDF). These results indicate that supervised methods may select features that can distinguish target vs. background data, but which do not capture more subtle phenomena within the target data.
>
> This extended analysis will be included in our revised manuscript.
>
> **Similar language to [37]**: We thank the reviewer for pointing out this issue. These sentences providing basic background info on CA were included when drafting early versions of the manuscript and accidentally remained in our submission. They will be rewritten and/or given proper credit in the final version. Thank you again for pointing out this issue!
>
> [1]: Jones, A., et al. Contrastive latent variable modeling with application to case-control sequencing experiments
>
> [2]: Weinberger, E., Lin, C. & Lee, SI. Isolating salient variations of interest in single-cell data with contrastiveVI
>
> [3]: Aglinskas, A. et al., Contrastive machine learning reveals the structure of neuroanatomical variation within autism
>
> [4]: Haber, A.L., et al. A single-cell survey of the small intestinal epithelium

---

> > ### Comment · Reviewer_xS67 · 2023-08-14
> > **Response to authors**
> >
> > I thank the authors for responding to all my comments. My concerns have been properly addressed. I keep my score unchanged.

---

> > > ### Author Response · Authors · 2023-08-15
> > > **Thank you for engaging in the discussion period!**
> > >
> > > Dear Reviewer xS67,
> > >
> > > Thank you for engaging in the discussion period. We're thrilled to hear that your concerns have been addressed! Please let us know if you have any further questions before the end of the discussion period.

---

### Official Review · Reviewer_BGi6 · 2023-07-06

**Soundness:** 3 good
**Presentation:** 3 good
**Contribution:** 2 fair
**Rating:** 5
**Confidence:** 3

**Summary:**

They employed a contrastive analysis method for feature selection on the target dataset, which provides variation not found in the background dataset. Contrary to previous contrastive analysis that typically involves joint modelling of background and target datasets to identify shared variables, they drew from information-theoretic analysis to demonstrate that a two-step pre-training model outperforms joint modelling.

**Strengths:**

Feature selection is a common requirement in health data to enhance interpretability. This paper clearly explained why the method proposed by them has a better performance than previous joint modelling and have good downstream accuracy when the number of features is limited.

**Weaknesses:**

That method needs strong assumptions that may not hold in the real world. Assumption one states that the latent variables s, z are roughly independent, but the genes in scRNA-seq (usage case in its real data experiments) are mostly highly correlated.

**Questions:**

In the real dataset, I noticed that all classes are larger than 1. What happens if there's only one class? Can the method still find variation that distinguishes it from the background? Regarding AML treatment, if the goal is to identify features that separate post-treatment from pre-treatment, could we use pre-treatment as the background and consider the selected features from post-treatment as the ones that have changed? Alternatively, would the method identify the same features as the current settings, which require healthy patients for the background?

As for scenarios where the number of features exceeds 30, do all methods eventually achieve similar accuracy? From Figure 5, I noted that STG performs similarly to CFS in the Mice Protein and Epithelial Cells datasets, though it is slightly worse in the AML and ECCITE_seq datasets. Could you provide information on the accuracy when the number of features is increased to 50?



**Limitations:**

Same as above weakness. Hard to satisfy the assumption that independent features.

---

> ### Author Rebuttal · Authors · 2023-08-09
>
> We thank the reviewer for carefully examining our work and providing their feedback. We’re thrilled to hear that the reviewer found that our manuscript “clearly explained” why CFS achieved superior performance compared to previously proposed feature selection methods. We have responded to general concerns shared across reviewers in our main response, and in the space below we respond to the reviewer’s specific concerns.
>
> **Clarifying the scope of our contribution:** The reviewer noted in their summary that a major contribution of our manuscript was demonstrating (both empirically and via our information-theoretic analysis) that our proposed two-step training procedure can lead to improved segregation of shared and target-specific variations, and we gratefully thank the reviewer for acknowledging this contribution.
>
> However, we also wanted to clarify that the contribution of our manuscript goes beyond just proposing the two-step training procedure. Indeed, to our knowledge, our manuscript is the **first work to consider the problem of feature selection in the contrastive analysis setting**, and our proposed CFS architecture is the first model specifically designed to take advantage of the weak supervision available in this scenario
>
> We will make this point more clear in a revised version of the manuscript.
>
> **AML/ECCITE-seq results for more than 30 features:** As requested by the reviewer, we performed additional experiments assessing the performance of CFS and baseline methods for >30 features on the AML and ECCITE-seq experiments. Below we provide results for $k=40, 50$ features as well as $k=30$ for comparison. We found that CFS continued to exhibit superior performance compared to baseline methods for the AML and ECCITE-seq datasets, though the magnitude of the difference between CFS and the best-performing baseline methods did narrow as the number of features increased.
>
> AML:
>
> | $k$ |CAE (U) | DUFS (U) | STG (S) | LassoNet (S) | CFS-Joint (C) | CFS-Pretrained (C) | CFS-SG (C) |
> |:---:|:---:|:---:|:---:|:---:|:---:|:---:|:---:|
> | 30 | $0.934 \pm 0.004$ | $0.688 \pm 0.016$ | $0.943 \pm 0.002$ | $0.669 \pm 0.003$ | $0.932 \pm 0.003$ | $0.965 \pm 0.002$ | $\boldsymbol{0.972 \pm 0.001}$ |
> | 40 | $0.938 \pm 0.002$ | $0.722 \pm 0.017$ | $0.925 \pm 0.008$ | $0.956 \pm 0.002$ | $0.932 \pm 0.003$ | $0.971 \pm 0.000$ | $\boldsymbol{0.973 \pm 0.000}$ |
> | 50 | $0.952 \pm 0.003$ | $0.754 \pm 0.015$ | $0.942 \pm 0.003$ | $0.964 \pm 0.001$ | $0.932 \pm 0.003$ | $\boldsymbol{0.976 \pm 0.001}$ | $0.972 \pm 0.001$ |
>
> ECCITE-seq:
>
> | $k$ | CAE (U) | DUFS (U) | STG (S) | LassoNet (S) | CFS-Joint (C) | CFS-Pretrained (C) | CFS-SG (C) |
> |:---:|:---:|:---:|:---:|:---:|:---:|:---:|:---:|
> | 30 | $0.857 \pm 0.002$ | $0.743 \pm 0.015$ | $0.860 \pm 0.002$ | $0.835 \pm 0.015$ | $0.849 \pm 0.001$ | $\boldsymbol{0.879 \pm 0.002}$ | $0.874 \pm 0.004$ |
> | 40 | $0.862 \pm 0.003$ | $0.746 \pm 0.014$ | $0.867 \pm 0.003$ | $0.866 \pm 0.010$ | $0.855 \pm 0.003$ | $0.881 \pm 0.001$ | $\boldsymbol{0.883 \pm 0.002}$ |
> | 50 | $0.866 \pm 0.002$ | $0.749 \pm 0.014$ | $0.870 \pm 0.003$ | $0.869 \pm 0.007$ | $0.858 \pm 0.000$ | $\boldsymbol{0.882 \pm 0.002}$ | $0.879 \pm 0.003$ |
>
>
> **What happens with only one class in the target dataset?:** We thank the reviewer for raising this point and for allowing us to clarify some potential points of confusion. First, we want to emphasize that CFS is designed for weakly supervised scenarios where we know only whether a data point is from the target or background dataset (so detailed class labels and/or a specific number of classes are not available). The goal of CFS is then to select features that best reflect target-specific variations without the aid of more detailed labels. Such a capability is valuable for analyzing many forms of biomedical data. For example, when analyzing large-scale single-cell genetic knockout perturbation screens, we may wish to identify features with a wide variety of responses to different perturbations (and which could subsequently be used to categorize the individual perturbations) without detailed knowledge ahead of time of how the various perturbations affect each feature.
>
> To accomplish this, CFS selects features that are most helpful for reconstructing target points given their low-dimensional background representations. To validate CFS’s performance, we assessed how well the features selected by CFS could distinguish between known ground truth classes of target points. However, when there is only one ground truth class of target points, the reconstruction objective will still encourage it to select features that exhibit variations unique to the target dataset (and which may distinguish target vs. background data). To illustrate this point, we trained CFS models on a modified version of the Grassy MNIST dataset, in which the target dataset consisted solely of images with the “1” digit. We found (**Fig. 3** in the rebuttal PDF) that CFS indeed selected features only in foreground regions where the “1” may be found. Similarly, if, as suggested by the reviewer, pre-treatment cells from the AML dataset were used as a background with post-treatment cells as the target, CFS would likely select features that distinguish pre- versus post-treatment cells.

---

### Author Rebuttal · Authors · 2023-08-09

We thank the reviewers for their thoughtful comments. We’re thrilled to hear that reviewers found CFS a “novel attempt at feature selection in a weak supervision setting”, “new and highly appropriate” (xS67), and “with practical interest” (nNUw). We’re also thrilled to hear that reviewers found our paper “well-written” (nNUw) with “strong” motivation (nNUw), and that our method was “clearly explained” (BGi6).

We respond to to the reviewers’ comments individually, and we also highlight our responses to some shared concerns below. Overall, we have made a number of helpful additions to the paper and we hope you will consider raising your scores.

**Clarification of architectural details (xS67, nNUw):**

For all experiments, the background representation encoder $g$ was implemented as a multilayer perceptron with input size equal to the number of features, one hidden layer, and an output dimension specified by the hyperparameter $l$ (the “background representation dimension”). The background reconstruction networks for the pretrained variant of CFS (i.e., $h$) were implemented with the same architecture in reverse.

The target reconstruction function $f$ was implemented as a network with two hidden layers. As inputs, the target reconstruction function $f$ took the concatenated background representation $\mathbf{b}$ and selected target features $x_{S}$, with output dimension equal to the original number of features.

We used ReLU activation functions for all the networks. For CFS, CAE, and STG, the sizes of all hidden layers were set to 512 for experiments with MNIST and 128 for the tabular biomedical datasets, as done in previous work with the CAE [1]. Due to the large number of models being trained, we did not tune the hidden layer size.


**Clarifying the role of the assumptions in Section 5 (BGi6, nNUw):**

In Assumption 1 of the analysis in Section 5, we assume that the salient and shared latent variables ($\mathbf{s}$ and $\mathbf{z}$, respectively) are roughly independent. As noted by the reviewers, this assumption may not hold in realistic datasets. For example, variations in gene expression relating to a cell's response to a given pathogen (i.e., target-specific salient variations governed by $\mathbf{s}$), such as those considered in the epithelial cell dataset from Haber et al. [2] used in our manuscript, may depend on that cell’s cell type (governed by the shared latent variables $\mathbf{z}$).

We thus emphasize that this assumption is not a requirement for applying CFS; rather, this assumption was made to aid in our analysis of the joint training procedures proposed in previous works on contrastive analysis versus our newly proposed two-step procedure. Indeed, with the aid of this simplifying assumption, we were able to prove why the two-step approach may lead to superior performance, as well as uncover a previously unappreciated potential failure case of previous joint contrastive training strategies.

This finding was empirically validated first on our semi-synthetic Grassy MNIST dataset, where we knew a priori that the salient and shared latent factors were fully independent, and our two-step procedure consistently outperformed joint training. Similarly, we found that our two-step procedure continued to outperform joint training on the four real-world biomedical datasets where it was not guaranteed that $\mathbf{s}$ and $\mathbf{z}$ would be independent. These additional empirical results thus suggest that the conclusions from our information-theoretic analysis may continue to hold even in cases when $\mathbf{s}$ and $\mathbf{z}$ exhibit realistic, nontrivial dependence.

**Additional ablations (SoSQ, nNUw):**

Based on the reviewers’ suggestions, we added additional ablation studies to better understand how CFS’s performance varies with respect to the method’s hyperparameters, and the properties of the target and background datasets. In particular, we

* Further experimented with the impact of varying the background dimension $l$ for different numbers of selected features $k$
* Assessed how performance varied with changes in the size of the background dataset versus the target dataset
* Assessed the impact on performance of randomly initializing the background representation encoder $g$ without training it on any background data.

We refer the reviewers to our responses to individual reviews for more details.


**In-depth analysis of the features selected by CFS on real-world data (xS67, nNUw)**:

To illustrate how CFS may aid in uncover new biological insights, we performed further analysis on the features selected by CFS for the mice intestinal epithelial cell infection dataset from Haber et al. [2]. In short, we found that CFS uncovered cell-type-specific changes in gene expression for a subset of genes in response to *Salmonella* infection that were not discussed in the original study. This finding thus illustrates the potential of CFS to lead to new biological insights that may be overlooked by standard analysis workflows. Due to space constraints we have placed the full details of this analysis in our individual response to reviewer **xS67**.

[1]: Covert, I., et al. "Learning to maximize mutual information for dynamic feature selection." International Conference on Machine Learning. PMLR, 2023.

[2]: Haber, A.L., et al. "A single-cell survey of the small intestinal epithelium."

---

### Decision · Program_Chairs · 2023-09-21

**Decision:**

Accept (poster)

**Comment:**

My recommendation is to accept this paper.

The paper proposes a method for doing feature selection in contrastive analysis, which is widely used in some applied fields, e.g., biomedical. Confident reviewers noted the potential broad utility of the method. I would encourage the authors to incorporate reviewer comments on additional results and paper organization in the camera ready.